

# Updraft dynamics and microphysics: on the added value of the cumulus thermal reference frame in simulations of aerosol-deep convection interactions

Daniel Hernandez-Deckers[1], Toshihisa Matsui[2], and Ann M. Fridlind[3]

[1]Grupo de Investigación en Ciencias Atmosféricas, Departamento de Geociencias, Universidad Nacional de Colombia, Bogotá, Colombia
[2]NASA Goddard Space Flight Center, Greenbelt, USA
[3]NASA Goddard Institute for Space Studies, New York, USA

**Correspondence:** Daniel Hernandez-Deckers (dhernandezd@unal.edu.co)

**Abstract.** One fundamental question about atmospheric moist convection processes that remains debated is whether or under what conditions a relevant variability in background aerosol concentrations may have a significant dynamical impact on convective clouds and their associated precipitation. Furthermore, current climate models must parameterize both the microphysical and the cumulus convection processes, but this is usually implemented separately, whereas in nature there is a strong coupling
between them. As a first step to improve our understanding of these two problems, we investigate how aerosol concentrations modify key properties of updrafts in eight large-eddy permitting regional simulations of a case study of scattered convection over Houston, Texas, in which convection is explicitly simulated and microphysical processes are parameterized. Dynamical and liquid-phase microphysical responses are investigated using two different reference frames: static cloudy-updraft grid cells versus tracked cumulus thermals. In both frameworks we observe the expected microphysical responses to higher aerosol
concentrations, such as higher cloud number concentrations and lower rain number concentrations. In terms of the dynamical responses, both frameworks indicate weak impacts of varying aerosol concentrations relative to the noise between simulations over the observationally derived range of aerosol variability for this case study. On the other hand, results suggest that analysis of thermals can provide a better pathway to sample the most relevant convective processes. For instance, vertical velocity from thermals is significantly higher at upper levels than when sampled from cloudy-updraft grid points, and several micro-
physical variables have higher average values in the cumulus thermal framework than in the cloudy-updraft framework. In addition, the thermal analysis is seen to add rich quantitative information about the rates and covariability of microphysical processes spatially and throughout tracked thermal lifecycles, which can serve as a stronger foundation for improving subgrid-scale parameterizations. These results suggest that cumulus thermals are more realistic dynamical building blocks of cumulus convection, acting as natural cloud chambers for microphysical processes.

## 1   Introduction

The net impacts of atmospheric aerosol concentration on deep convective cloud systems and their environment remain highly uncertain, with mixed results that do not generally yield conclusive answers yet (e.g., Khain et al., 2008; Tao et al., 2012). All



else being equal, a higher aerosol concentration generally corresponds to more condensation nuclei at any given supersatura-
tion, which in turn is expected to produce more but smaller cloud droplets within a convective updraft. This may delay the
occurrence of initial warm-precipitation formation due to a less efficient collision-coalescence process, enhancing latent heat
release above the freezing level (Rosenfeld et al., 2008). However, when or if this has substantial impact on the amount or in-
tensity of cold precipitation is not clear due to the uncertainties of subsequent ice and mixed-phase microphysics (e.g., Korolev
et al., 2020), and the complex morphology and feedback of deep convective clouds under various environmental conditions
(e.g., Tao et al., 2012; Fan et al., 2016; Abbott and Cronin, 2021). One approach to reduce such complexities to some degree
is to focus on aerosol-cloud interactions in relatively isolated convective cells (e.g., Fridlind et al., 2019), where the various
mechanisms by which aerosol may impact updraft properties remain operative.

The recent Aerosol-Cloud-Precipitation-Climate working group (ACPC) Model Intercomparison Project (MIP) compared
regional model simulations of such scattered convection in response to a realistic dynamic range of ambient aerosol concentra-
tion profiles with similar large-scale forcing. Although participating models exhibited similar updraft invigoration at low levels,
differences between models are larger than each model's response to ambient aerosol loading (Marinescu et al., 2021), offering
little clear guidance for larger-scale models. Indeed, it is even more challenging to represent such processes in a climate model,
because updraft microphysics and dynamics are often simplified by cumulus parameterization at much coarser spatio-temporal
resolution (McFarlane, 2011). To better represent such processes in climate models, it is imperative to disentangle aerosol-deep
convection interactions from the wider spectrum of microphysics and dynamical processes.

One foundational step in order to tackle this problem is to investigate the possible links between the updraft and micro-
physical processes in moist convection. Characterizing dynamical and microphysical properties in response to the ambient
aerosols is very difficult from existing observations, but current high resolution numerical models in which the dynamics of
convection are resolved, such as those analyzed by Marinescu et al. (2021) or Abbott and Cronin (2021), offer a useful alter-
native. In order to study convective cloud properties in such simulations, the "active" cloudy regions must be identified first,
which is traditionally done by sampling grid points with specific thresholds of vertical velocity and liquid water content; we
call these "cloudy-updraft grid points". Such active cloud-sampling criteria have been widely used since large eddy simula-
tions (LES) have been available (e.g., Siebesma and Cuijpers, 1995; de Roode and Bretherton, 2003). However, it is clear that
moist convection generally constitutes a series of many short-lived thermals within each cumulus cloud (Scorer and Ludlam,
1953; Woodward, 1959; Blyth et al., 2005; Damiani et al., 2006; Sherwood et al., 2013; Yano, 2014; Hernandez-Deckers and
Sherwood, 2016; Morrison and Peters, 2018), raising the question whether the traditional grid-point selection criteria are the
most appropriate. For instance, cumulus thermals themselves can be very heterogeneous due to their own internal circulation
structure (Hernandez-Deckers and Sherwood, 2016), so that traditional grid point sampling may miss relevant air masses.
In addition, traditional grid point sampling may include rising or cloudy points that are unrelated to the relevant convective
air masses (e.g., Mrowiec et al., 2015). This can be avoided with even more selective criteria, such as that by Marinescu
et al. (2021), who only include grid points within 6 km deep (or more) cloudy-updraft columns, thus considering only well-
developed deep convective cores. However, important microphysical activity may also occur outside of such cores, and their





initial lifetime stages remain unaccounted for. All this suggests the possibility of a more dynamically-based definition of the active cloudy regions arising from cumulus thermals.

Identification and tracking of cumulus thermals in numerical simulations has been used to investigate their intrinsic dynami-
cal properties in studies such as those by Sherwood et al. (2013), Romps and Charn (2015), Hernandez-Deckers and Sherwood (2016), Hernandez-Deckers and Sherwood (2018), Moser and Lasher-Trapp (2017), Lecoanet and Jeevanjee (2019), or Peters et al. (2020). Their results have contributed to improve the understanding of the dynamical properties and the role of thermals in cumulus convection, which is necessary for the development of new convection parameterization schemes. However, to our knowledge cumulus thermal identification has not been used as a sampling approach similar to the traditional cloudy-updraft
grid points or convective core identification. Here, we apply the thermal identification and tracking method of Hernandez-Deckers and Sherwood (2016) to use it as a novel sampling approach, and compare it to the traditional cloudy-updraft grid point method in the context of dynamical and microphysical impacts on deep convection due to changes in aerosol concentrations.

The more complex cumulus thermal framework has the potential to enable insights regarding the coupling between updraft
dynamics and microphysics, which are not visible from the traditional cloudy-updraft grid point framework. For example, we find that thermals act as natural cloud chambers, with microphysical processes mostly contained within them and driven by their internal circulations. Both frameworks are expected to provide important information about the impact of aerosol concentrations on the dynamical and microphysical properties of deep convection. However, given that thermals are the basic dynamical entities of cumulus clouds, their response to different aerosol concentrations provides a fundamental link to the
overall convective response. Although the ultimate aerosol impact on precipitation amount and intensity may depend on details of the particular microphysical parameterizations used, the first step we carry out here is to use both reference frames to investigate the basic impacts on the initial warm-phase microphysics and dynamics within scattered isolated convection. Through a series of relatively high-resolution large-eddy permitting regional model experiments, this study investigates the impact of a sequential increase in aerosol concentrations on the simulated dynamics and microphysics of deep convection.
From the microphysical point of view, we focus on warm-phase microphysics, because of larger uncertainties in ice nucleation and subsequent ice and mixed-phase microphysics.

Following this introduction, section 2 describes the simulations analyzed here, as well as a summary of the thermal identification and tracking method. Section 3 presents the main results, first in terms of composites of thermals, next in terms of vertical profiles of various quantities, and finally comparing the cloudy-updraft grid point and thermal frameworks. Section 4
presents the summary and conclusions of this study.

## 2 Simulations and methods

### 2.1 Case study and model set up

The case study is based on scattered, isolated convective clouds that developed over Houston, Texas on 19-20 June 2013, following the ACPC MIP simulations (Marinescu et al., 2021). During daytime, the heating over land, especially urban regions,





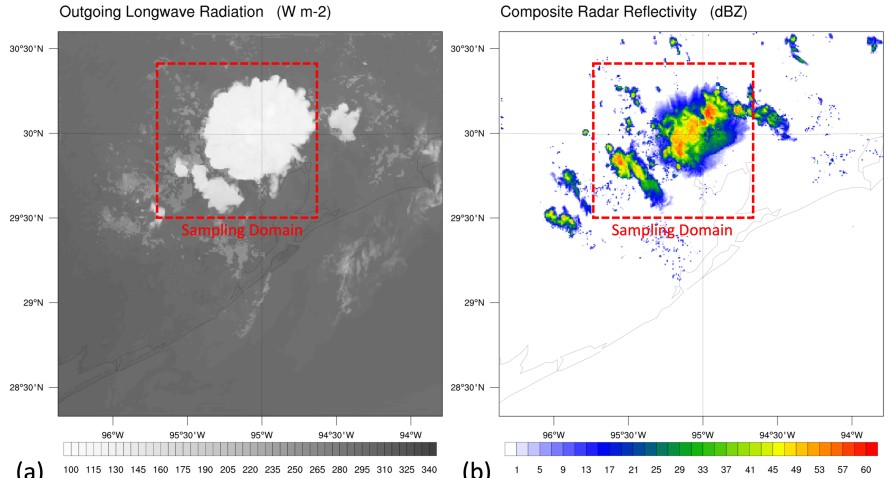

**Figure 1.** NU-WRF domain and sampling domain showing (a) Outgoing Longwave Radiation (OLR) and (b) composite radar reflectivity on 19 June 2013 at 23:25 UTC, for the simulation with an aerosol number concentration of 500 cm$^{-3}$.

develops a pressure gradient between land and ocean. The associated afternoon sea-breeze front triggers scattered convection by disturbing conditionally unstable layers. This study uses the NASA-Unified Weather Research and Forecasting (NU-WRF) model (Peters-Lidard et al., 2015) configuration that was also used in the ACPC MIP study as a basis (Marinescu et al., 2021); however, the domains, grid spacing, and aerosol concentrations are revised in order to investigate cumulus thermals.

This case study utilizes a single large domain (998x998 horizontal grid cells) with 250 m horizontal grid spacing, without
nesting (Fig. 1). Analysis is focused on scattered convection that occurs due to mesoscale circulations within the domain. Vertical grid spacing stretches from approximately 50 m near the surface to 300 m near the 4 km level with 96 vertical levels. The planetary boundary layer (PBL) parameterization was turned off, and only the turbulent kinetic energy (TKE) scheme is used; we found that the TKE scheme with the PBL scheme at this resolution unphysically suppresses the number of cumulus thermals within the middle of boundary layer (not shown). Other physics options include the new Goddard radiation scheme
(Matsui et al., 2020), NOAH-MP land surface model, and Predicted Particle Properties (P3) scheme with a single ice species.

The P3 scheme predicts mass and number concentrations of cloud droplets, rain drops, and ice particles, and additional tracers (rime mass and volume) are also predicted to better characterize ice properties (Morrison and Milbrandt, 2015). Aerosol activation follows Abdul-Razzak and Ghan (2000) using minimum supersaturation from Morrison and Grabowski (2008) (their Eq. A10). Based on regional observations (Marinescu et al., 2021), aerosol profiles spanning the boundary layer (up to 2500 m
above ground level) are stratified in the eight sensitivity experiments from relatively clean continental (500 cm$^{-3}$) up to polluted conditions (4000 cm$^{-3}$), increasing by 500 cm$^{-3}$ for each sensitivity experiment. Aerosol is specified as a single-mode lognormal distribution with fixed mean diameter (100 nm), lognormal distribution width (1.8), and hygroscopicity parameter (0.2). As in Marinescu et al. (2021), aerosol transport (resolved and sub-grid), activation, removal by droplet coalescence, and regeneration from droplet evaporation follows the method in Fridlind et al. (2017). NCEP Final Analysis (FNL) was used to



initialize NU-WRF on 19 June 2013 at 12:00 UTC and continued updating lateral boundary conditions until 20 June 2013 at 15:00 UTC. Since thermal tracking requires 1-min temporal resolution of NU-WRF output, we focused on the three-hour time window from 19 June 21:00 UTC for thermal and cloudy-updraft grid point analysis during the active convection period. Figure 1 shows the actual sampling domain used, where most active convection occurs during this time window.

## 2.2  Thermal identification and tracking

Sufficiently high-resolution simulations can generally reproduce the observed thermal-like structures that are characteristic of cumulus clouds (e.g., Sherwood et al., 2013; Varble et al., 2014; Romps and Charn, 2015). This provides a numerical tool to investigate the dynamics of these thermals, which in turn leads to a better understanding of many aspects of convection (Morrison, 2016; Moser and Lasher-Trapp, 2017; Hernandez-Deckers and Sherwood, 2018; Peters et al., 2020). Here we identify, track and analyze cumulus thermals in the NU-WRF simulations described in the previous section using the methodology of

Hernandez-Deckers and Sherwood (2016). In the following we describe the main features of this method; for further details, please refer to their study.

To identify thermals, an automated algorithm identifies peaks in vertical velocity throughout a particular volume of the simulation at each output timestep. It then tracks each thermal's center location in time, and from the obtained trajectories it estimates each thermal's ascent rate. Assuming spherical shapes, a thermal's radius can be estimated at each timestep by

requiring that the average vertical velocity of the enclosed region matches the thermal's ascent rate. This ensures that each identified thermal corresponds to a coherent rising volume of air. Hernandez-Deckers and Sherwood (2016) showed that indeed thermal shapes do not deviate much from sphericity, making this a good approximation. Finally, it is worth noting that the algorithm only takes into account thermals with average ascent rates of at least $1\,\mathrm{m\,s^{-1}}$, and with centers that have at least $0.01\,\mathrm{g\,kg^{-1}}$ of cloud condensate. Furthermore, it computes each thermal's vertical momentum budget and discards any cases

in which the tracked trajectory is inconsistent with it. From the sample of tracked thermals, different statistical measures can be obtained for both microphysical and dynamical properties. These can be then compared to results based on the cloudy-updraft sampling framework. For consistency, our threshold criteria for selecting cloudy-updraft grid points is a vertical velocity of $1\,\mathrm{m\,s^{-1}}$ and a cloud condensate of $0.01\,\mathrm{g\,kg^{-1}}$.

The mass flux captured by the tracked thermals is typically 15-20% of the estimated total mass flux, as will be shown below.

Despite this being a relatively small fraction, Hernandez-Deckers and Sherwood (2016) showed that the convective evolution is well represented by the thermals, suggesting that their dynamics are representative of the entire convective activity (discussed later). Untracked updrafts are typically too small or too slow to be tracked with this algorithm. Furthermore, the total mass flux is not uniquely defined, and may contain spurious non-convective contributions (e.g., Mrowiec et al., 2015). Finally, it is worth noting that we find very similar properties of thermals in this study compared to what Hernandez-Deckers and Sherwood

(2016) found with their higher resolution simulations (65 m horizontal grid spacing). The only prominent difference is that our thermals are larger (R $\sim$1.2 km, compared to R $\sim$0.3 km), which may be expected given our coarser spatial resolution setting but could also be partially attributable to differences in the case study conditions. The resolution used here is within the range found to be adequate for resolving dynamics of continental deep convection (Bryan et al., 2003). Owing to the similarity





of results to Hernandez-Deckers and Sherwood (2016), we expect that finer resolution results would be more converged but
similar in nature.

## 3    Results

### 3.1    Thermal composites

Figure 2 shows statistical composites of microphysics properties within tracked thermals from the selected background aerosol
cases of 500, 1000, 2000, and 4000 $cm^{-3}$ (i.e., for each subsequent doubling of aerosol concentrations). The results demonstrate
that an increase in background aerosol concentrations tends to (a) increase cloud droplet nucleation rates, (b) reduce super-
saturation values, (c) increase cloud droplet number concentrations, and (d) decrease rain number concentrations (Figs. 2a-d).
Plots of average values of these quantities within thermals as a function of aerosol number concentration (not shown here)
reveal that nucleation rates, supersaturation values and cloud drop number concentration behave roughly linearly with aerosol
number concentration, whereas rain number concentration decreases exponentially, consistent with rain drop generation by
coalescence of cloud droplets. On the other hand, although number concentrations of both cloud droplets and raindrops are
strongly affected by aerosol number concentration, their mixing ratios respond less strongly (Figs. 2e-f) and in such a way that
the total liquid water mixing ratio remains more weakly impacted (not shown here).

Microphysical quantities are found to peak at thermal centers nearly universally, which reinforces the important role of
thermals as building blocks of convection from both a dynamical and microphysical point of view. For example, supersaturation
values are only reached inside thermals, generating numerous cloud droplets around their cores. Streamlines of the averaged
flow also indicate more turbulent mixing around the thermal frame, whereas upstream currents are present in the core of
thermals. As illustrated further below, each convective cloud contains many such natural cloud chambers (i.e., thermals).

The microphysical response to aerosol number concentration could cause a prominent dynamical response in thermals via
changes in the rate at which latent heat is released due to condensation. For example, following the reasoning by Fan et al.
(2018), a reduction in supersaturation rates could result from the larger number of smaller droplets (and hence more available
surface area for condensation) as aerosol concentrations increase. All else being equal, this could imply faster latent heat release
due to condensation. However, Fig. 3a indicates no prominent mean response in latent heating rates within the tracked thermals,
while cloud nucleation rates increase and supersaturation rates decrease with increasing aerosol concentrations (Figs. 2a-b).
This implies that there is no prominent change in diabatic heating per unit time available for the dynamics of the thermals,
which indicates similar total condensation rates despite changes in driving supersaturations. Figures 3b-c also show no notable
changes in their composite buoyancy (B) or vertical velocity (w). Although not shown here, we do not find any prominent
trends in terms of the thermals' composite lifetime, vertical distance traveled or radius. Furthermore, the histograms of these
quantities are negligibly changed (not shown).

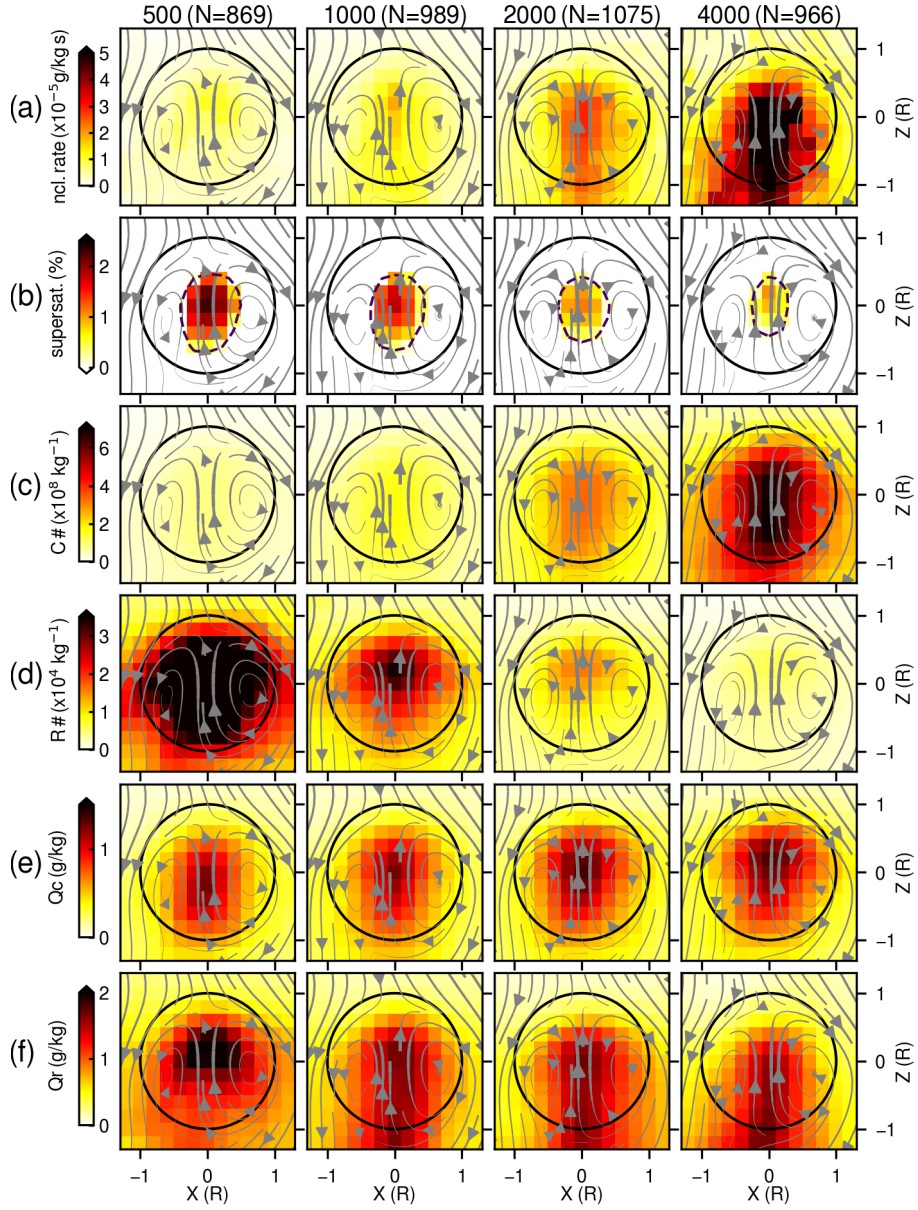

**Figure 2.** Cross sections along the xz plane of mean values of (a) cloud nucleation rate, (b) supersaturation values, (c) cloud drop number concentration C#, (d) rain number concentration R#, (e) cloud liquid water mixing ratio Qc, and (f) rain mixing ratio Qr, for composites of all tracked thermals scaled by their radius (horizontal and vertical coordinates are in units of mean thermal radii). Each column corresponds to a simulation with initial aerosol number concentration (indicated above, in $cm^{-3}$). N corresponds to the number of tracked thermals used for the composites. Arrows indicate streamlines of the average flow in the reference frame of the rising thermal. The dashed contour in supersaturation values corresponds to 100% relative humidity.

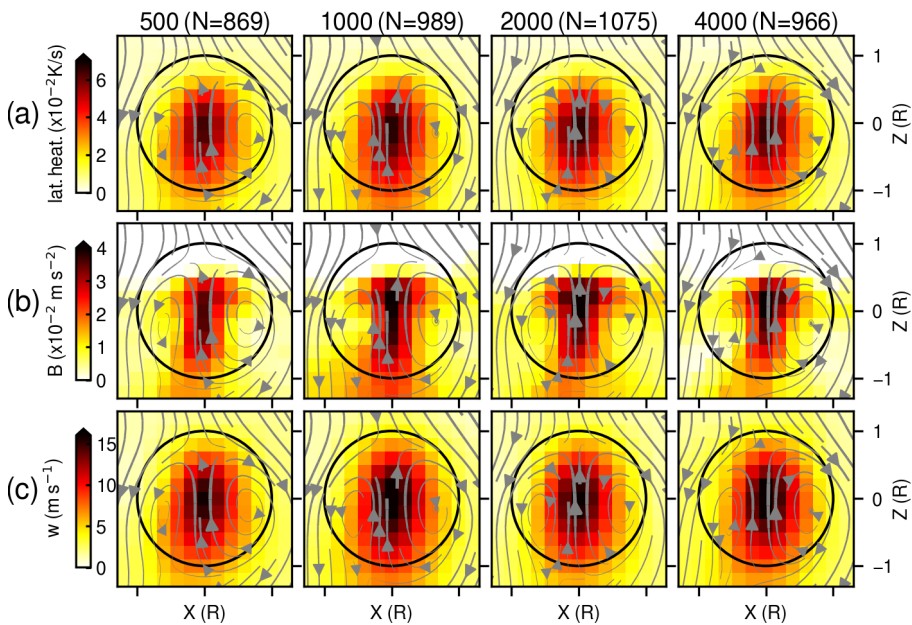

**Figure 3.** Composites for all tracked thermals as in Fig. 2, but for (a) latent heating rates, (b) buoyancy, and (c) vertical velocity.

## 3.2 Vertical profiles

Since many of these variables have strong vertical dependencies, we next investigate these responses in terms of vertical profiles of microphysical quantities, latent heating rates, vertical velocity, and mass flux, as estimated from cloudy-updraft grid points (Figs. 4a-h) and from the tracked thermals (Figs. 4i-p). To begin with, notice that vertical profiles in both frameworks show qualitatively consistent features at most elevations. Perhaps the most prominent difference between these two frameworks is that thermals indicate a larger contribution than cloudy-updraft grid points to several quantities at levels above 6-7 km

above ground level (AGL). This is very clear in terms of vertical velocity (Figs. 4g and 4o), where both frameworks yield very similar profiles up to ∼6-7 km AGL, but significantly different values aloft. According to cloudy-updraft grid points, vertical velocity reaches its maximum near 7 km AGL, whereas according to thermals it continues to increase, reaching its maximum near 10 km AGL. This suggests that the thermal sampling criteria is more selective of vigorous updrafts aloft. This also results in a slightly more top-heaviness of profiles of other quantities, highlighting the importance of such dynamically

relevant structures for microphysical processes. In terms of mass flux, both frameworks yield a maximum near 3 km AGL, but thermals indicate a secondary maximum between 7 and 9 km AGL. Notice that this corresponds to the contribution of relatively few thermals (Fig. 6a), suggesting that, unlike near cloud base where convection results from small contributions of many updrafts, convection at mid and high levels near cloud top is more tightly linked to the contribution of relatively few but vigorous updrafts, a feature that may be better captured by the cumulus thermal framework.





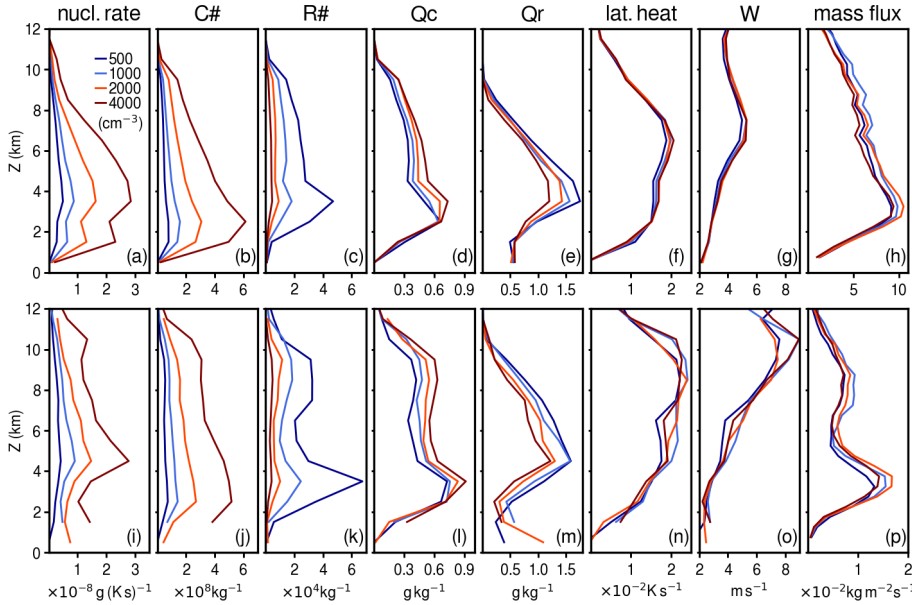

**Figure 4.** Vertical profiles of cloud nucleation rate (a), (i), cloud number concentration (b), (j), rain number concentration (c), (k), cloud water mixing ratio (d), (l), rain water mixing ratio (e), (m), latent heating rates (f), (n), vertical velocity (g), (o), and mass flux (h), (p), for experiments with different aerosol number concentrations (see legend in panel (a)). Top panels (a)-(h) are computed from cloudy-updraft grid points, and lower panels (i)-(p) from tracked thermals. Notice the different scales used for mass flux in panels (h) and (p).

Regarding the responses to increases in aerosol concentrations, both frameworks show overall agreement. To visualize these responses more clearly, Figure 5 shows the differences between profiles for each successive doubling of aerosol concentrations, their average change, and the difference between the most and least polluted cases. This figure also helps to identify in which quantities there is a consistent response to increases in aerosol concentrations. Notice how linear the response is for cloud nucleation rate and cloud droplet number concentration (Figs. 5a-b and 5i-j), and how the decrease in rain number concentration

behaves exponentially, with the largest changes for lower aerosol number concentrations (Figs. 5c and 5k). On the other hand, the increase in cloud water mixing ratio is slightly offset by the decrease in rain water mixing ratio (Figs. 5d-e and 5l-m), so that there is a slight net decrease in total liquid water mixing ratio (not shown here). However, the variability in the decrease in rain water mixing ratio between pairs of experiments is significantly higher than in the increase in cloud water mixing ratio, which also makes the net decrease in total water mass highly variable between simulations.

In contrast to the microphysical quantities, latent heating rates, vertical velocity and mass flux do not reveal such prominent and consistent responses to aerosol concentrations (Figs. 5f-h and 5n-p), and this is true in both frameworks. As expected, changes in vertical velocity closely follow changes in latent heating rates, but both are small on average, with a high level of noise between different pairs of experiments, more so in the cumulus thermal framework. For example, in the comparison between 4000 and 500 cm$^{-3}$ we find an increase of $\sim$10% in vertical velocity near heights of 6 and 11 km AGL (consistent





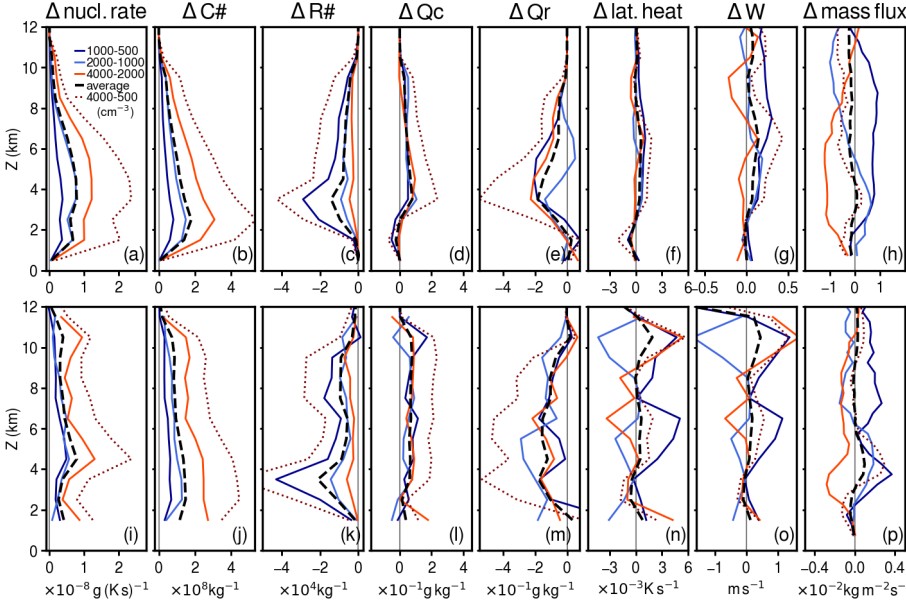

**Figure 5.** Differences between vertical profiles in Fig. 4, corresponding to each doubling of aerosol number concentrations (continuous colored lines), their average change (dashed black line), and the change between the two extreme cases, 4000 and 500 cm$^{-3}$ (dotted line). Top panels (a)-(h) correspond to cloudy-updraft grid points, and lower panels (i)-(p) to tracked thermals. Notice the different scales used for vertical velocity and for mass flux.

with findings by Marinescu et al., 2021). The average response for a doubling of aerosol concentrations at these altitudes also suggests an increase, but much weaker ($\sim$2%); however, not all individual pairs of cases show such an increase, and the amplitude of the individual responses is usually larger than the average one.

Regarding mass flux, notice that its estimate based on tracked thermals is $\sim$15% of the cloudy-updraft estimate (Figs. 4h,p). As shown by Hernandez-Deckers and Sherwood (2016), the relatively low fraction captured by thermals results from mainly
small and slow thermals that are harder to identify and track with our method; however, it is representative of the total mass flux. In fact, notice that the changes in mass flux for each doubling are consistent between the thermal-estimate and the cloudy-updraft grid point estimate (Figs. 5h,p). Here too, the average response for doubling aerosol concentrations is weaker than individual responses. In fact, it is nearly zero everywhere, except for a slight increase around 4 km AGL in the thermals' framework, which can be linked to an increase in the number of tracked thermals (Fig. 6e).
Similar results are seen for other quantities relevant for cumulus thermals (Fig. 6). A certain degree of correspondence can be seen between buoyancy changes (Fig. 6g) and vertical velocity changes (Fig. 5o), with hardly any average response when doubling aerosol concentrations despite significant (but not consistent) changes between individual pairs of simulations. Changes in the average vertical distance traveled by thermals (DZ, Fig. 6h), is also similar to changes in vertical velocity of

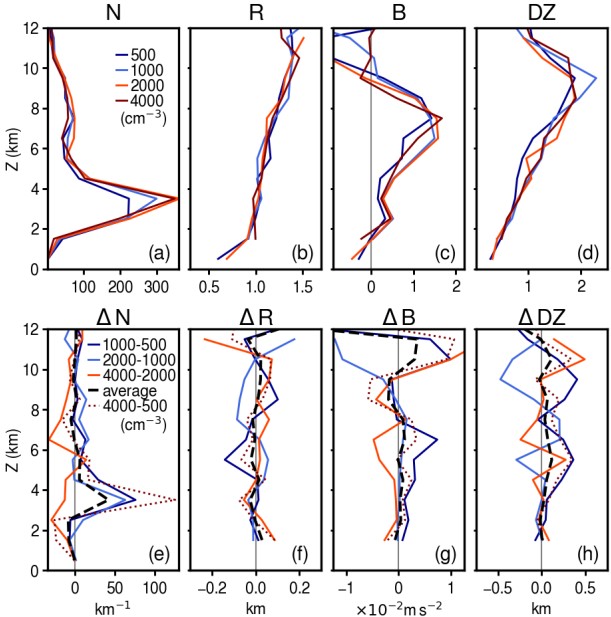

**Figure 6.** Vertical profiles of (a) number of thermals (per vertical km), (b) average thermal radius, (c) average buoyancy, and (d) average vertical distance traveled by thermals, for the different aerosol number concentrations (see legend). Panels (e) through (h) show the differences in the quantities of panels (a) through (d) between successive pairs of profiles (continuous colored lines), their average change (dashed black line), and the change between the two extreme cases, 4000 and 500 cm$^{-3}$ (dotted color line).

thermals, especially its average response for a doubling of aerosol concentrations (Fig. 5o). This indicates that average thermal
lifetime (not shown here) is also invariant to aerosol number concentrations.

All these quantities related to the thermals' dynamics seem to respond only very weakly to changes in aerosol number concentrations, compared to the natural variability between each pair of simulations. This is a known limitation when investigating aerosol invigoration of convection. Several studies have emphasized the difficulty of rising above the 'noise level' when trying to identify aerosol indirect effects (e.g., Morrison and Grabowski, 2011; Grabowski, 2014). For a given microphysics
and dynamics framework, our results support this view from both the cloudy-updraft and the thermal frameworks regarding fundamental dynamical properties, since results vary widely depending on which pair of experiments is taken into account. However, we also see some indication of a change in the sign of the trend across the full dynamic range of aerosol variability. For instance, doubling aerosol initially increases buoyancy near 6 km AGL, but ultimately decreases it at that elevation by a similar amount when reaching the highest aerosol concentration. Similar responses can be seen in terms of w, DZ, and mass
flux, consistent with an "aerosol-limited" regime (e.g., Koren et al., 2014).

Average thermal size, which we estimate here with its radius R, shows no systematic change related to aerosol number concentrations (Figs. 6b and 6f). However, we do find a response in the number of tracked thermals, particularly between 2-4 km AGL, where most thermals develop. This response also seems to depend on the particular range of aerosol variability, with





more thermals being tracked as aerosol concentrations increase in the "cleaner" regime (500-2000 $cm^{-3}$), and fewer thermals
being tracked when aerosol concentrations are doubled in the more polluted regime (2000-4000 $cm^{-3}$).

### 3.3 Thermals vs. cloudy-updraft grid points

We have shown how our two sampling criteria provide a general agreement in terms of the microphysical and dynamical
responses to increases in aerosol number concentrations. However, we have also noted differences, which reveal important
features of thermal and grid point analyses. The scatter plots in Fig. 7 show how relevant quantities averaged within thermals
compare to the same quantities averaged over cloudy-updraft grid points, both for different vertical layers (circle dots) and
for the entire columns (crosses) in the different aerosol number concentration experiments. In general, these plots confirm that
both thermal and cloudy grid points analyses are close to each other, but interesting features emerge from their comparison.

Cloud and rain number concentration as well as cloud mass mixing ratio (Figs. 7a and 7e) appear to be similar between
thermal and cloudy grid points, but have slightly higher values within thermals than for cloudy-updraft grid points; this is
more prominent at higher altitudes, where thermals tend to be larger and vigorous, and so as for rain number concentrations,
suggesting that thermals indeed act as natural cloud chambers. In other words, at higher elevations, thermals differ more
from the "average" cloudy conditions than at lower elevations, which emphasizes their important role in the deepening of the
convective cloud. At near surface level (~1 km AGL), cloud number and mass concentrations are lower than the cloudy-updraft
grid points, most likely due to the thermal's internal circulations that may include downdrafts and/or condensate-free volumes
of air, but nevertheless are dynamically connected to the rising thermals and their internal microphysical processes.

Rain mass mixing ratios also appear to be higher in thermals than in cloudy-updraft grid points, but have on average similar
values in both cases (Fig. 7f). When separated by height, thermals at higher altitudes tend to have higher rain mass mixing
ratios than cloudy-updraft grid points, but the opposite is true at lower altitudes. This can be explained if one thinks of thermals
at upper levels as the regions where rain is starting to form—and hence have more rain mass than the average cloudy-updraft
grid points, whereas rain at lower levels tends to be concentrated at downdraft regions, where rising thermals are limited. An
interesting feature here is that the average values per experiment cross the 1:1 line in such a way that thermals have higher rain
mixing ratios than the average cloudy updraft grid points in the cleaner cases, but lower rain mixing ratios in the polluted cases.
This would be in line with raindrops being larger (and fewer) in the polluted cases, making them fall faster and less likely to
be inside a rising thermal.

Averaged over the entire vertical column, thermals and cloudy-updraft grid points respond almost equally in terms of nu-
cleation rates to varying aerosol number concentrations (Fig. 7c). However, thermals tend to have slightly higher nucleation
rates in the upper levels, and lower nucleation rates in the lower levels compared to cloudy-updraft grid points. This small
difference may be because thermals in the upper levels tend to sample the larger, faster, and hence less diluted updrafts, while
the cloudy-updraft grid points may also sample weaker, shorter-lived updrafts, where nucleation rates are lower. On the other
hand, at lower altitudes thermals tend to be smaller and more numerous, likely sampling similar updrafts as cloudy-updraft grid
points, but thermals include a larger volume of air surrounding the updrafts, slightly reducing their average nucleation rates.



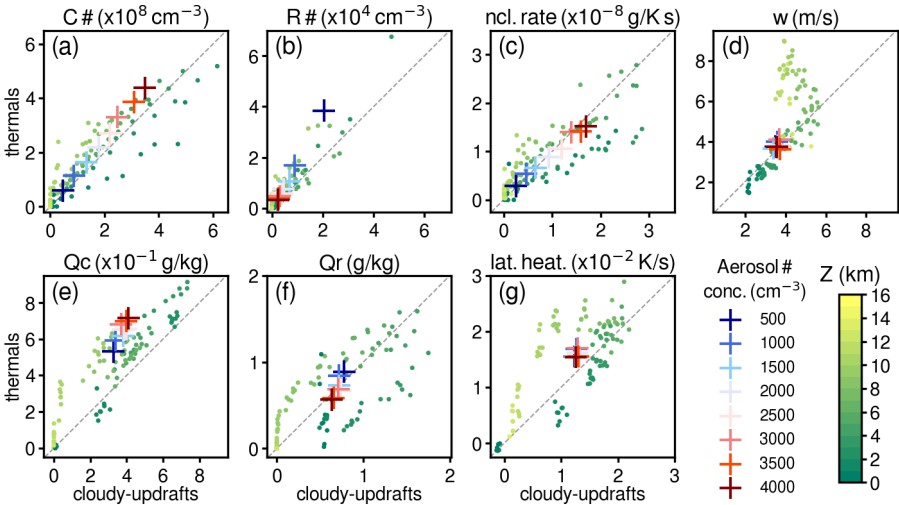

**Figure 7.** Scatter plots of (a) cloud drop number concentrations C#, (b) rain number concentrations R#, (c) cloud nucleation rates, (d) vertical velocity w, (e) cloud water mixing ratio Qc, (f) rain water mixing ratio Qr, and (g) latent heating as obtained from averaging over thermals (vertical axis) and over cloudy-updraft grid points (horizontal axis). Averages over thermals are computed obtaining first an average value for each thermal, and then averaging over all thermals at a certain altitude range (color dots), or averaging over all thermals (crosses, with colors according to aerosol number concentration of each experiment). Values for cloudy-updraft grid points are obtained by averaging these per altitude range (color dots), or per experiment (crosses).

In terms of overall column averages, we see that for both thermals and cloudy-updraft grid points, latent heating and vertical velocity appears to be similar (Figs. 7d and 7g). Regarding the relation between thermal averages and cloudy-updraft grid points, there are important differences with altitude. For example, the average vertical velocity of cloudy-updraft grid points and thermals follows the 1:1 line closely up to about $6\,\mathrm{m\,s^{-1}}$. Average vertical velocity for thermals, in particular above an altitude of about 6 km AGL, does exceed this value, while the average for cloudy-updraft grid points does not. To understand this, notice that the mass flux captured by thermals (Fig. 4e) has a first maximum just below 4 km AGL, and a second maximum around 8-9 km AGL. The first maximum coincides with the layer where most smaller and short-lived thermals are found within the boundary layer; the second maximum has about half the mass flux of the first, but only about a sixth of the number of thermals (Fig. 6a). Thus, the thermals above 6-7 km AGL are not as numerous, but larger ones individually contribute much more to the total mass flux than those in the boundary layer.

Finally, the fact that latent heating rates tend to be higher for thermals than for cloudy-updraft grid points at higher altitude (Fig. 7g) suggests that thermals are capturing the most relevant regions where condensation occurs and thus the most relevant convective regions of the cloud. Latent heating rates of thermals largely exceed those of cloudy-updraft grid points at higher altitude, but underestimate at near surface level. These are very similar patterns of those combined from cloud and rain mass mixing ratio (Figs. 7e-g). Overall, these results highlight the important role of thermals in convective microphysics, dynamics and mass flux, which could be at least partially eclipsed when analyzing only cloudy-updraft grid points.





## 4 Summary and conclusions

In order to investigate the coupling between updraft dynamics and microphysics, we study the impact of ambient aerosol
concentration on deep convection in a series of eight simulations at 250 m horizontal grid spacing of a case study over Houston,
Texas, where initial background aerosol concentrations are systematically varied from $500\,cm^{-3}$ to $4000\,cm^{-3}$ in intervals of
$500\,cm^{-3}$. Apart from the traditional cloudy-updraft grid point analysis (e.g., summarized in Tao et al., 2012; Fan et al., 2016),
we also identify and track cumulus thermals, and use these as an alternative sampling criteria to study the deep convective
response to the imposed aerosol concentrations, based on the idea that thermals are the building blocks of cumulus clouds
(e.g., Sherwood et al., 2013; Varble et al., 2014; Romps and Charn, 2015). Comparative analysis between cloudy-updraft grid
points and cumulus thermals provide new insights into the coupling between updraft dynamics and microphysics.

As a first step, and given the uncertainties in the current representation of convective microphysical processes, this study
focuses only on the warm phase microphysics. We find similar microphysical responses to an increase in aerosol concentrations,
for thermals and for cloudy-updraft grid point analyses: nucleation rates and cloud drop number concentrations increase, while
supersaturation values and rain number concentrations decrease. That is, more—but smaller—cloud droplets form, leading to
fewer—but larger—rain drops. These responses are very consistent throughout the entire sets of experiments, indicating a clear
connection to aerosol number concentrations in rising thermals, and cloudy-updraft grid points. However, average latent heating
rates are not impacted by changing aerosol concentrations, except in the middle troposphere (4 and 6 km AGL), where average
~2% increases of latent heat rates, ascent rate and vertical velocity occur for every doubling of aerosol number concentrations,
similarly between thermal and cloudy-updraft grid point analyses.

Nevertheless, these responses for thermals and cloudy-updraft grid points are not entirely consistent between individual pairs
of doubling experiments. Thus, very different conclusions could be drawn from each pair of experiments due to natural vari-
ability (e.g., Morrison and Grabowski, 2011; Grabowski, 2014) and several other factors, such as the synoptic forcing, ambient
relative humidity, the actual range of aerosol concentrations, and specific microphysics schemes (Fan et al., 2007; White et al.,
2017; Barthlott and Hoose, 2018; Iguchi et al., 2020; Abbott and Cronin, 2021; Marinescu et al., 2021). Therefore, results of
this type are usually case- and model-dependent, and conclusions from a single model configuration or a single—or few—
cases should be interpreted with caution. Our simulations, which intend to replicate a real continental case where only aerosol
number concentrations are varied over an observationally established range, suggest that natural variability largely surpasses
the impact of aerosols on the dynamical features of convection. It is therefore not surprising that inter-model variability has
also been found to be larger than aerosol-related variability in terms of its impact on convection (e.g., Marinescu et al., 2021).

Despite the uncertainties of the model response to background aerosol concentrations, the comparison between cloudy-
updraft grid points and thermals reinforces the idea that thermals act not only as dynamical building blocks of convective clouds,
but also microphysical building blocks as natural cloud chambers. Thermals, especially in the middle and upper troposphere,
are larger, more vigorous, and undiluted so that they nucleate higher droplet and raindrop concentrations, and higher cloud
water mixing ratios than the average cloudy-updraft grid points, acting as rain incubators, too. On the other hand, at the lower
troposphere (below 4 km AGL), where smaller short-lived thermals are predominant, updraft velocity, cloud nucleation and



latent heating rate of thermals tend to be equivalent or smaller than cloudy-updraft grid points, likely due to the thermals' internal heterogeneity, which may also be important to consider. Consequently, thermal microphysics quantities tend to be also equivalent or lower in thermals than in cloudy-updraft grid points at such altitudes. This suggests that thermals and cloudy-updraft grid points are similar sampling criteria in the lower troposphere, but from the middle troposphere upward, large and vigorous thermals may offer a more selective sampling criteria that captures the most relevant convective air masses, where microphysical processes are indeed most active. However, the thermal tracking approach yields an abundance of additional information on the spatiotemporal evolution and lifecycle of the cloud chambers that largely drive hydrometeor production processes within convective clouds; indeed, this is the key information needed for subgrid-scale parameterizations in climate models and into the grey zone in which convective processes remain poorly resolved. Overall, this further motivates the use of thermals as the basic elements to develop a parameterization of coupled convective dynamics and microphysics for a climate model to better represent aerosol-deep convection interactions in the future.

*Code availability.* The NASA-Unified WRF is maintained at NASA GSFC, and available for public release upon request (https://nuwrf.gsfc.nasa.gov/). NU-WRF outputs are available upon request via NASA GSFC Cloud Library (https://portal.nccs.nasa.gov/cloudlibrary/). All other processing code used for this study is available upon request to the authors.

*Author contributions.* All authors conceived and designed the research, TM carried out the model simulations and performed the cloudy-updraft grid point analyses, DH-D performed the thermal tracking and its analysis and led the manuscript writing with input from all authors.

*Competing interests.* The authors declare no conflict of interest.

*Acknowledgements.* We thank the NASA Advanced Supercomputing (NAS) Division for providing the computational resources to conduct and analyze the NU-WRF simulations. TM is funded by DOE ASR program (DE-SC0021247) and NASA PMM program (Grant number: 80NSSC19K0724), AMF was supported by the Office of Science (BER), U.S. Department of Energy, under Agreements DE-SC0006988 and DE-SC0016237, and DH-D is funded by Universidad Nacional de Colombia.





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
