# Peer review of "Updraft dynamics and microphysics: on the added value of the cumulus thermal reference frame in simulations of aerosol-deep convection interactions"

_Atmospheric Chemistry and Physics, 2021_

## Referee Comment (RC1)

**Hernandez-Deckers et al., Updraft dynamics and microphysics: on the added value of the cumulus thermal reference frame in simulations of aerosol-deep convection interactions**

This paper uses three hours from a borderline-resolution large-eddy simulation of convection over Houston to analyze differences between results from two approaches to sampling cloud elements. The first method, called the *cloudy-updraft* method, identifies the active portions of clouds as the regions where there is coincident cloud condensate and upward motion. The second method, referred to as the *tracked thermals* method, is more complicated and uses a tracking algorithm to look for peaks in the vertical velocity field, infer the associated thermal bubbles, and then track them over time. Each method is applied to a series of simulations that differ solely by the aerosol concentration used with the cloud microphysics. The results show that the overall story about aerosol-cloud processes is similar between the two approaches, but an apparent sampling bias leads to subtle differences. The use of a range of aerosol concentrations also highlights the difficulty of separating signal from noise in aerosol-cloud analyses.

Overall, the paper is well presented—it is both organized well and the English is very clean. The science is sound and the comparison between methods is an important investigation that informs how researchers can use and intercompare results from these methods in the future. This is where the primary value lies in this paper, as the findings for the aerosol-cloud processes essentially mirror what has been found in prior studies.

This paper will be suitable for publication after addressing a small number of questions and suggestions noted below.

**Specific Comments**

(1) Intro.    The introduction builds from the ACPC MIP in terms of indicating that the differences between models are typically larger than the sensitivity to aerosol details within a model. This makes it hard to then infer true process rates and details for use in subsequent parameterization development, etc. The argument is then made that one should disentangle the dynamical details of the deep convection from the microphysical processes. The implication is that the bubble tracking procedure achieves this. However, this misses the points that the model-to-model differences still exist, and many of the microphysical factors within the thermals rely upon the dynamics, such as the supersaturation being dependent on the vertical velocity. More importantly, if the present study were repeated using a different model, e.g., that treats autoconversion and other processes differently, the results could substantially change, such as happened between models in the MIP. There is likely a better way to frame the introduction to be more useful for the study that is presented.

(2)    Even though this set of simulations is based on a realistic setup, the presentation of the results essentially assumes this is an idealized setup. There is no reference to an observation anywhere in the paper in relation to the simulation or results. This

paper would be strengthened by putting the aerosol and meteorological state in the context of reality. Is the simulation anywhere close to the observed conditions for this day? Which aerosol concentration is closest to reality?

(3) L96       Please state the model top, as this is relevant in relation to both the number of levels and the height of the convection.

(4) L109–10       I have some concern about the use of a single domain with such high resolution driven only by the FNL. Assuming the 0.5° FNL data set was used, that is roughly a factor 200 jump in grid spacings along the lateral boundaries. How much more accurate would the results be if a more traditional nested approach were used to step down to $\Delta x=250$ m from the $\Delta x=0.5°$ of the FNL?

(5) L109–10       Given the non-steady-state nature of the land-sea breeze driving the convection, what is the impact of the infrequent boundary updates on the convection? The lack of nesting would likely exacerbate this issue. Please state the frequency of the boundary condition updates (likely 6 h).

(6) L124–5       The phrasing about the thermal radius was not immediately clear without referring back to the Sherwood (2013) paper for clarification. Please reword this to add a little more detail. For example, my impression the first couple times I read this sentence was that there was some sort of scaling relationship being used, which is not the case (a misconception that stuck in my head from reading too fast the first time). It is clearer now that I understand what is being done, but it took me a little digging to make myself confident I understood how the radius is calculated. Adding a little more detail will help readers out.

(7) L139–41       This sentence infers that the thermal/bubble structures identified both in this study with 250 m grid spacing and in Hernandez-Deckers and Sherwood (2016) with 65 m grid spacing are a strong function of the model numerics and not a resolved feature. Both studies are just identifying features at WRF's effective resolution—dynamical structures finer than this would tend to not be coherent. Thus, the actual behavior of the thermals in nature could be somewhat different than what is being seen in the simulations. While I realize it is not feasible to use a model domain that has converged results for the identified cloud features, one should at least note this limitation to the reader.

(8) Figs. 2 & 3       Visually the presentation of the composited thermal characteristics in Figs. 2 & 3 is quite effective in conveying how the aerosol concentrations impact the cloud state within the thermals. Having a bit more description about how the composites were constructed would help interpretation. Most importantly, what portions of the thermal lifetimes are averaged together? Would it be more useful to look at certain times within the lifecycle, such as at maximum translation velocity or at a given altitude? Otherwise, conditions with low and high cloud water concentrations are averaged together and could hide important features.

(9) Figs. 4–6       The profiles in Figs. 4–6 are presumably averaged over the three analyzed hours. Is there any evolution of the clouds during this time? For example, how far inland has the land-sea breeze moved? Does this impact the results at all?

(10)     The authors speculate that the differences identified between the cloudy-updraft and tracking methods is due to sampling bias. However, this is not confirmed beyond pointing out a physically consistent argument that sounds plausible. Is there a way to alter the cloudy-updraft method to reduce the selection bias and confirm the speculation? For example, can one add an additional criterion such that the vertical velocity must be within the values that are consistent with the velocities seen in the thermals identified via tracking? Finding a way to make the comparison fairer and seeing if the differences between methods go away would greatly strengthen the paper.

---

## Referee Comment (RC3)

**Review of "*Updraft dynamics and microphysics: on the added value of the cumulus thermal reference frame in simulations of aerosol-deep convection interactions*"**

Model-based analysis of aerosol indirect effects, or other properties of convection in general, typically relies on some definition of an updraft. Often this is done by considering all cloudy points in a model with some threshold vertical velocity value. The authors here use object tracking code to follow individual thermals, offering this as an alternative to the cloudy grid points method. They use this method to investigate some simple aerosol indirect effects in simulations of deep convection. Generally the aerosol effects on the warm part of the storms are as expected, and are fairly consistent between the two methods, but some differences are seen in the upper levels that suggest the thermal tracking method could be a useful way to investigate processes in deep convection

Overall I think the paper is interesting, novel, and sound, but I offer some comments and suggestions below to help improve and clarify the discussions.

Comments:

It seems to me that the authors overstate the importance of their method compared to the traditional approach. There is no "correct" answer in how to do the analysis. Both methods may prove useful for examining different characteristics or different types or regions of storms. It's especially concerning to me that the authors concentrate on the differences in the upper levels, yet this study doesn't at all investigate the ice phase. It's difficult to conclude anything about microphysics in the upper levels of a storm if only cloud and rain are included in the analysis. This merits some mention at least. What implications might the differences between these two approaches offer when examining the ice phase?

Line 170: It doesn't seem consistent that there could be huge increases in nucleation rates, but not in latent heating. Is this being balanced by evaporation?

Line 172: I think it would be better to show at least some of this information. I found myself wondering about the properties of the thermals and the variability of those properties at multiple times while reading, and was frustrated to keep seeing "not shown".

Line 188: While it is likely true that the *convective core* is more tightly linked at upper levels to fewer, stronger updrafts, the fact remains that the additional updrafts not captured as thermals do exist there, and are certainly quite relevant for microphysics. By only looking at the strongest updrafts in the upper levels, it may better capture that core, but is ignoring stratiform processes. It's not clear that one of these methods would be better than the other in general - it depends on the question being asked.

Line 210: I'm not sure that I buy that it's representative of the total mass flux. In figure 4 you show that the profiles have different shapes as well as the difference in magnitude. They are

not really looking at the same thing, which is your point, so how is one representative of the other?

Figure 6: How are there 300 individual thermals in this fairly small domain?

Figure 6: With 250m grid spacing, an average radius of 500m doesn't make a lot of sense. These thermals are barely resolved, and this being the average means you are also tracking thermals that are fewer than 4 points across.

---

## Author Comment (AC1)

[Figure]

Fig. S1: As in Fig. 4, but for the early stage of the simulation, between 21:00 UTC and 21:45 UTC.

[Figure]

Fig. S2: As in Fig. 4, but for the middle stage of the simulation, between 22:00 UTC and 22:45 UTC.

[Figure]

Fig. S3: As in Fig. 4, but for the later stage of the simulation, between 23:00 UTC and 23:45 UTC.

[Figure]

Fig. S4: As in Fig. 7, but using w>2 m/s for the cloudy updraft grid point definition.

[Figure]

Fig. S5: As in Fig. 7, but using w>4 m/s for the cloudy updraft grid point definition.

---

## Author Response (AR1)

**Reply to Anonymous Referee #1 of the manuscript**

**"Updraft dynamics and microphysics: on the added value of the cumulus thermal reference frame in simulations of aerosol-deep convection interactions"**

by
Daniel Hernandez-Deckers, Toshihisa Matsui, and Ann M. Fridlind

In the following we provide a point by point response to comments by Reviewer #1, where we quote in italics the original comments. We have numbered all comments in bold face, based on the reviewer number and comment number so that the different replies can be easily referred to within the text (e.g. **R2C3** refers to comment number 3 from Reviewer #2).

*"This paper uses three hours from a borderline-resolution large-eddy simulation of convection over Houston to analyze differences between results from two approaches to sampling cloud elements. The first method, called the cloudy-updraft method, identifies the active portions of clouds as the regions where there is coincident cloud condensate and upward motion. The second method, referred to as the tracked thermals method, is more complicated and uses a tracking algorithm to look for peaks in the vertical velocity field, infer the associated thermal bubbles, and then track them over time. Each method is applied to a series of simulations that differ solely by the aerosol concentration used with the cloud microphysics. The results show that the overall story about aerosol-cloud processes is similar between the two approaches, but an apparent sampling bias leads to subtle differences. The use of a range of aerosol concentrations also highlights the difficulty of separating signal from noise in aerosol-cloud analyses.*

*Overall, the paper is well presented—it is both organized well and the English is very clean. The science is sound and the comparison between methods is an important investigation that informs how researchers can use and intercompare results from these methods in the future. This is where the primary value lies in this paper, as the findings for the aerosol-cloud processes essentially mirror what has been found in prior studies.*

*This paper will be suitable for publication after addressing a small number of questions and suggestions noted below."*

We thank the reviewer for this summary and reply in the following lines to each specific comment:

*"Specific Comments*
**R1C1**: *(1) Intro.*
*The introduction builds from the ACPC MIP in terms of indicating that the differences between models are typically larger than the sensitivity to aerosol details within a model. This makes it hard to then infer true process rates and details for use in subsequent parameterization development, etc. The argument is then made that one should disentangle the dynamical details of the deep convection from the microphysical processes. The implication is that the bubble tracking procedure achieves this. However, this misses the*

*points that the model-to-model differences still exist, and many of the microphysical factors within the thermals rely upon the dynamics, such as the supersaturation being dependent on the vertical velocity. More importantly, if the present study were repeated using a different model, e.g., that treats autoconversion and other processes differently, the results could substantially change, such as happened between models in the MIP. There is likely a better way to frame the introduction to be more useful for the study that is presented."*

Indeed, one limitation of this study is that results are based on a single model, and it is important to mention this more explicitly. We have added now the following text to the introduction starting at line 81 (lines 87-91 of track changes version):

> "Here we investigate the dynamics/microphysics coupling using a single model and case study with two analysis approaches; because differences between both models and case studies are expected (e.g., Tao et al., 2012; Marinescu et al., 2021), however, it will not be possible to establish the generality of our results to other models and scenarios without future work, whose potential merit may nonetheless be in part guided by our initial findings here."

**R1C2:** *"(2) Even though this set of simulations is based on a realistic setup, the presentation of the results essentially assumes this is an idealized setup. There is no reference to an observation anywhere in the paper in relation to the simulation or results. This paper would be strengthened by putting the aerosol and meteorological state in the context of reality. Is the simulation anywhere close to the observed conditions for this day? Which aerosol concentration is closest to reality?"*

We have added the following sentence starting at line 109 (lines 123-126 of the track changes version):

> "The polluted and clean aerosol size distributions and vertical profiles were based on the data from Deriving Information on Surface conditions from Column and Vertically Resolved Observations Relevant to Air Quality (DISCOVER-AQ) in September 2013, as well as satellite-based estimates (Rosenfeld et al. 2012) near Houston on 19 June 2013. Timing of satellite CCN observations are identical to the simulation dates."

**R1C3:** *"(3) L96*
*Please state the model top, as this is relevant in relation to both the number of levels and the height of the convection."*
We have added the following sentence at line 97 (line 109 of the track changes version):
> "Model top is approximately 22 km (50 hPa)."

**R1C4:** *"(4) L109–10*
*I have some concern about the use of a single domain with such high resolution driven only by the FNL. Assuming the 0.5° FNL data set was used, that is roughly a factor 200 jump in grid spacings along the lateral boundaries. How much more accurate would the results be if a more traditional nested approach were used to step down to Δx=250 m from the Δx=0.5° of the FNL?"*

This is a case-specific semi-idealized simulation. Obviously, the model downscaling in these experiments far exceeds the traditional stretching ratio (3:1~5:1), because our intention is to create thermal bubbles of isolated convection driven by sea-breeze circulation (Fig. 1).  In

particular, NU-WRF have used quilting and compressed IO options in WRF, which asynchronously passes model output memory to a set of computational nodes for dumping WRF output with compression format (sizes of 1000x1000x90 grids) at one-minute intervals. While this option enables us to generate a number of LESs with limited computational resources, this often crashes with multiple nested domain options.

Fig. R1 shows that the single-domain simulation tends to invigorate isolated convection and is more concentrated over the Houston area later in time than in the triple domain simulation. It also misses the convection in the northeast of the simulation domain in comparison with the radar observation (Fig. R2).

Certainly, forecasting skill can be degraded in the single domain. But our purpose is to generate sea-breeze-driven isolated convection for a given set of land-ocean contrast in surface temperature and pressure with mean flow from boundary conditions. Thus, the single domain set is convenient and enough for our science objective.

We added the following sentence starting at line 95 (lines 105-107 of the track changes version):
> "This type of domain setting exceeds the traditional downscaling ratio (1:3~1:5), resulting in reduced precipitation forecasting skill compared to multi-nested domains. However, it successfully generates thermal bubbles of isolated convection driven by sea-breeze circulation for a given computational resource."

[Figure]

Fig R1: Time series (21Z, 22Z, 23Z, and 24Z) of total precipitable water (blue shade), near-surface wind vector, and composite radar reflectivity from 250m-mesh single domain simulations used in this study (top panel) versus 500m-mesh triple domains (bottom panel) used in Marinescu et al. (2021).

[Figure]

Fig R2. Observed radar composites from NEXRAD from Marinescu et al. (2021).

**R1C5:** "*(5) L109–10*

*Given the non-steady-state nature of the land-sea breeze driving the convection, what is the impact of the infrequent boundary updates on the convection? The lack of nesting would likely exacerbate this issue. Please state the frequency of the boundary condition updates (likely 6h)."*

Boundary conditions are compiled at 6 hourly intervals, but actually it is interpolated in time and space to force the model at every model time step. The simulated sea-breeze circulation is driven by land-ocean contrast in temperature and pressure gradients within the domain. This sea-breeze dynamics was explicitly simulated within the domain. We have added the following clarification starting at line 111 (lines 135-137 of the track changes version):

> "Six-hourly lateral boundary conditions from GFS are spatially and temporally interpolated to update the model lateral boundary conditions at every model time step, while sea-breeze dynamics are explicitly simulated by model physics and dynamics within the domain."

**R1C6:** "*(6) L124–5*

*The phrasing about the thermal radius was not immediately clear without referring back to the Sherwood (2013) paper for clarification. Please reword this to add a little more detail. For example, my impression the first couple times I read this sentence was that there was some sort of scaling relationship being used, which is not the case (a misconception that stuck in my head from reading too fast the first time). It is clearer now that I understand what is being done, but it took me a little digging to make myself confident I understood how the radius is calculated. Adding a little more detail will help readers out."*

We have rephrased this explanation (lines 122-125) in order to make it as clear as possible, but without providing unnecessary details that would end up reproducing the methodology description given by Hernandez-Deckers and Sherwood (2016). The improved text corresponds to lines 149-154 in the track changes version:

> "To identify thermals, an automated algorithm identifies peaks in vertical velocity throughout a particular volume of the simulation at each output timestep, and assumes that these indicate the instantaneous locations of thermals' centers. By comparing these locations in consecutive output timesteps, the algorithm can estimate each thermal's trajectory, which also yields an estimate of their ascent rates at each timestep. Assuming spherical shapes, a thermal's size can be estimated by

choosing the radius that makes the average vertical velocity of the enclosed volume match the corresponding ascent rate. Notice that each thermal's ascent rate can vary between timesteps, and hence the estimated size of a thermal may also vary in time."

**R1C7:** *"(7) L139–41*
*This sentence infers that the thermal/bubble structures identified both in this study with 250 m grid spacing and in Hernandez-Deckers and Sherwood (2016) with 65 m grid spacing are a strong function of the model numerics and not a resolved feature. Both studies are just identifying features at WRF's effective resolution—dynamical structures finer than this would tend to not be coherent. Thus, the actual behavior of the thermals in nature could be somewhat different than what is being seen in the simulations. While I realize it is not feasible to use a model domain that has converged results for the identified cloud features, one should at least note this limitation to the reader."*

It is true that average thermal size seems to be strongly influenced by the model's spatial resolution, which raises the question whether the resolution of the model is good enough. On the one hand this implies that one should be careful not to take these simulated thermal sizes as the exact "real" ones in nature. On the other hand, the fact that their dynamical properties are similar at both resolutions is reassuring and suggests that we can expect these to also hold in nature, as already stated in lines 143-145:

> "Owing to the similarity of results to Hernandez-Deckers and Sherwood (2016), we expect that finer resolution results would be more converged but similar in nature."

**R1C8:** *"(8) Figs. 2 & 3*
*Visually the presentation of the composited thermal characteristics in Figs. 2 & 3 is quite effective in conveying how the aerosol concentrations impact the cloud state within the thermals. Having a bit more description about how the composites were constructed would help interpretation. Most importantly, what portions of the thermal lifetimes are averaged together? Would it be more useful to look at certain times within the lifecycle, such as at maximum translation velocity or at a given altitude? Otherwise, conditions with low and high cloud water concentrations are averaged together and could hide important features."*

Composites are constructed using only one "instant" of each thermal's lifecycle. The "instant" chosen for each thermal is that in which the thermal has its highest ascent rate, i.e., when it is most vigorous. We now mention this in the main text after line 149 (lines 179-180 in the track changes version):

> "For these composites, only the timestep of maximum ascent rate of each thermal is considered."

However, as the reviewer points out, this could smooth out certain features by averaging over different stages of the thermals' lifecycle. Figure R3 shows the same composites of Fig. 2, but based on the instant 3 minutes before (left panel) and 3 minutes after (right panel) each thermal reaches its maximum ascent rate.

The only new feature that arises here is that rain and cloud water mixing ratios are higher during the earlier stages of the thermal than during the later stages, consistent with liquid water content increasing during the initial intensifying stages of thermals and decreasing as thermals decay, as expected. However, we find no new features related to the different aerosol concentrations.

[Figure]

Fig. R3: Composites as in Fig. 2 of the manuscript, but sampling thermals 3 minutes before (left) and 3 minutes after (right) they reach their maximum ascent rate.

Furthermore, Fig. R4 shows the composites corresponding to Fig. 3 in the manuscript, (latent heating rates, buoyancy and vertical velocity), but 3 minutes before and after thermals reach their maximum ascent rate. Once again, the only feature is that these quantities have all higher values during the early stages of thermals, and lower during the later stages. No new feature related to aerosol concentrations is seen here either.

[Figure]

Fig. R4: Composites as in Fig. 3 of the manuscript, but sampling thermals 3 minutes before (left) and 3 minutes after (right) they reach their maximum ascent rate.

Finally, creating composites by altitude could be interesting, but for that to be really informative, a much larger sample of thermals at different altitudes would be required. However, the vertical profiles in Figs. 4-6, as well as the scatter plots in Fig. 7, already provide useful information in this respect.

**R1C9:** *"(9) Figs. 4–6*
*The profiles in Figs. 4–6 are presumably averaged over the three analyzed hours. Is there*
*any evolution of the clouds during this time? For example, how far inland has the land-sea*
*breeze moved? Does this impact the results at all?"*

First of all, the sea-breeze penetrates inland, where we have set the sampling box (see Fig. R1), and as it does this, isolated convection becomes stronger and starts to aggregate over the Houston area. This may indeed make certain features of convection different throughout the 3 analyzed hours. However, since we are interested in the possible impacts of different aerosol concentrations upon convection in general, we believe it makes sense to consider the three hours together. Furthermore, by doing so we are able to obtain a larger sample of updrafts/thermals, reducing the noise level. However, we agree that it is also important to assess if these impacts, or in general the response of convection to aerosols is different throughout the different stages of convection.

In order to do this, we have constructed 3 versions of Figs. 4-6, each corresponding to a different stage of the simulation. The early stage starts at 21Z (same starting time of the full 3-hour period), the middle stage starts at 22Z and the final stage starts at 23Z (see Figures R5-R7 in the following pages). The resulting profiles are consistent with the time evolution of convection as it deepens. Mass flux increases at upper levels during the middle and final stages, and vertical velocity also increases aloft with time (Figs. R5A-C). This deepening is also very clear in the profiles of the number of thermals (Figs. R7Aa-R7Ca). An important point here is that due to the fact that these stages have fewer thermals (and updrafts) than when analyzing the full 3-hour period, the profiles are significantly noisier.

In terms of the responses to the different aerosol concentrations (Figs. R6A-C), we still see the same picture in the individual stages as over the entire three hours, but now with substantially more noise. One might argue that there are some differences in terms of how vertical velocity and latent heating rates respond to aerosols in the different stages. For example, during the early and late stages the average response in vertical velocity suggests a stronger invigoration, whereas during the middle stage the average response is rather a weakening. However, the individual responses of each doubling of aerosol concentrations are much more variable and noisy, so we argue that the averaged 3 hour-response is more robust. Furthermore, the responses in both frameworks are generally consistent with each other, but with more noise, particularly in the thermal framework.

We will now include Fig. R5 as supplement to the manuscript (as three separate figures), since we believe it provides some additional insight into the time-evolution of this convective case. However, Figs. R6 and R7 are too dominated by noise due to the short analysis time-interval, and thus do not justify additional supplementary material. Also, we have added the following paragraph after line 189 (lines 225-229 of the track changes version):

> "It is important to point out that throughout the 3-hour period analyzed here, convection evolves and may behave differently at different stages. To assess this, Figs. S1-S3 show profiles as in Fig. 4, where the three-hour period has been divided into three stages. These profiles reflect the fact that convection deepens with time, but otherwise show consistency with Fig. 4. Furthermore, considering the entire 3-hour period provides a larger sample of updrafts, which in turn aids in reducing the noise."

[Figure]

Fig. R5: As in Fig. 4 in the manuscript, but for 3 stages of the simulation, early (A), middle (B) and final (C).

[Figure]

Fig. R6: As in Fig. 5 in the manuscript, but for 3 stages of the simulation, early (A), middle (B) and final (C).

[Figure]

Fig. R7: As in Fig. 6 in the manuscript, but for 3 stages of the simulation, early (A), middle (B) and final (C).

**R1C10:** *"(10)*
*The authors speculate that the differences identified between the cloudy-updraft and tracking methods is due to sampling bias. However, this is not confirmed beyond pointing out a physically consistent argument that sounds plausible. Is there a way to alter the cloudy-updraft method to reduce the selection bias and confirm the speculation? For example, can one add an additional criterion such that the vertical velocity must be within the values that are consistent with the velocities seen in the thermals identified via tracking? Finding a way to make the comparison fairer and seeing if the differences between methods go away would greatly strengthen the paper."*

Actually, both frameworks have been implemented in the most consistent possible way, by using the same thresholds of vertical velocity and cloud condensate (see lines 132-133, or 162-163 in the track changes version). Changing these thresholds for the cloudy-updraft gridpoint framework would make them less consistent with each other. The difference that remains is really fundamental in terms of the dynamical coherence of the volume of air that is required for a thermal, something that is not possible to introduce in the cloudy-updraft gridpoint framework. However, following this suggestion, and a similar one by reviewer #2, we have tested two higher thresholds for vertical velocity in the cloudy updraft definition, and produced two figures as supplementary material (Figures S4 and S5). Please see reply to comment **R2C3.6** for the complete response and the corresponding additions after line 276 (lines 316-320 in the track changes version).

**Reply to Anonymous Referee #2 of the manuscript**

**"Updraft dynamics and microphysics: on the added value of the cumulus thermal reference frame in simulations of aerosol-deep convection interactions"**

by
Daniel Hernández-Deckers, Toshihisa Matsui, and Ann M. Fridlind

In the following we provide a point by point response to comments by Reviewer #2, where we quote in italics the original comments. We have numbered all comments in boldface, based on the reviewer number and comment number so that the different replies can be easily referred to within the text (e.g. R2C3 refers to comment number 3 from Reviewer #2).

*"Overview*

*This study applies cloudy updraft and tracked thermal frameworks to analyze updraft statistics in LES simulations of relatively isolated deep convection near Houston, TX. Sensitivity of updrafts to aerosol concentration between 500 and 4000 cm-3 is analyzed. Although cloud droplet and raindrop concentrations change significantly in response to aerosol changes, latent heating and vertical wind speed show little sensitivity. Buoyancy and thermal number sensitivities to aerosol concentration are non-monotonic. Both frameworks show similar sensitivities of updraft properties to aerosols. The primary difference is in the upper troposphere were the tracked thermal framework produces stronger updrafts. Magnitudes of effects also vary, but this is understandable given the two different sampling methods.*

*Overall, this is an interesting study comparing two different techniques that are commonly used for studying convective updrafts. I'm not aware of other such comparisons, which makes the results publishable. The aerosol sensitivities are also publishable, particularly since they disagree with many papers, some of which are case studies, that claim that increasing aerosol concentration increases convective vigor through the ice phase."*

**R2GC**: *"The primary issue with the study is that it stresses how the tracked thermal framework is superior to static cloudy updraft frameworks and that thermals are fundamental building blocks of convection that act as natural cloud chambers, but that is all very subjective without much evidence to support it. There are differences between the two framework results that make sense based on how they are sampling the model output. Despite that, they give results that are more similar than different with respect to aerosol sensitivities. Why one or the other is better connected to convective dynamics and microphysics understanding and parameterization is not clearly presented. Rather, it seems like each could be useful, particularly in providing context to each another and in supporting greater confidence in results when similar microphysical and dynamical sensitivities are similar in each, like seems to mostly be the case in this study. Without further results, it seems that this should instead be the message that is stressed most."*

Thank you for this insightful review of the manuscript. We agree that the main message should not be that the thermal framework is "better" than the cloudy-updraft grid point framework, but rather that both methods are consistent in most situations, and that by comparing them we can gain additional insights of updraft microphysics that are expected to be useful for understanding and parameterizing the coupling of aerosol and cloud dynamics. The way we had phrased several parts of the manuscript could be interpreted as the former

rather than the latter, so we made the following revisions (see the tracked changes version of the manuscript for comparing with the previous version):

Modified lines 12-13 (abstract) to: "...results suggest that thermals are more selective than cloudy-updraft grid cells in terms of sampling the most active convective air masses."

Modified lines 57-58 (lines 60-61 in the track changes version) to: "All this suggests the possibility of exploring an alternative, more objective-based definition of the active cloudy regions arising from cumulus thermals."

Modified text starting at line 69 (lines 73-79 in the track changes version) to: "The more complex cumulus thermal framework enables a direct, three-dimensional, structure-based analysis of how the internal updraft dynamical structure is coupled to the microphysical processes, something that is difficult to obtain from the grid point framework. Both frameworks are expected to provide important information about the impact of aerosol concentrations on the dynamical and microphysical properties of deep convection, and here we compare the approaches in a systematic fashion."

Modified lines 281-282 (lines 325-326 in the track changes version) to: "Overall, these results highlight how both frameworks are generally consistent, while subtle differences between them can potentially provide additional useful information."

Added text starting at line 322 (lines 368-370 in the track changes version): "This increases the level of noise in the thermal framework compared to the cloudy-updraft grid point framework, but that may also represent information content regarding the scarcity of what have sometimes been referred to as "lucky updrafts"."

Regarding lack of evidence that *"thermals are fundamental building blocks of convection that act as natural cloud chambers"*, we believe there are two main points here. First, the conceptual idea that thermals serve as the building blocks of cumulus clouds (or convection), something that has been put forward by many authors in the past (e.g., Zhao and Austin, 2005; Blyth et al., 2005; Damiani et al., 2006; Hernandez-Deckers and Sherwood, 2016). The main issue here, we believe, concerns the second point, which presents these thermals as "natural cloud chambers". The reviewer comes back to this point several times in the specific comments below. Indeed, thermals are not closed systems; in fact, as the reviewer points out, entrainment (and detrainment) is significant in typical thermals (e.g., Sherwood et al., 2013; Hernandez-Deckers and Sherwood, 2018; Lecoanet and Jeevanjee, 2019). Our intention is to offer a simple analogy for how a typical thermal's dynamical structure tends to concentrate most microphysical processes within it, without implying that they do not mix with their environment. This idea results from the composite plots in Figs. 2 and 3, and is certainly one of our key findings (even if it is only qualitative). It further provides the connection for improving subgrid-scale parameterizations (see specific comment below on this topic). This was our intention when using the term "natural cloud chamber", but we agree that this may not be the best analogy. We now avoid this term, and rather describe this feature more explicitly. This has led to the following changes:

- Removed the last sentence from the abstract (lines 19-20).
- Removed text from lines 69-72 (75-77 in the track changes version).
- Removed sentence from line 162 (193 in the track changes version).
- Removed text from line 246 (286 in the track changes version).
- Removed text from lines 312-313 (358-359 in the track changes version).
- Changed "cloud chambers" for "structures" at line 323 (line 372 in the track changes version).
- We also expanded discussion of potential advantages of the thermal approach for parameterization starting at line 325 (lines 374-379 in the track changes version):

"For instance, efforts to extend climate model convection schemes that parameterize updraft velocities and use these to inform microphysical process rates (e.g., Wu et al., 2009) can draw upon the three-dimensionally colocated properties and process statistics directly identified within the structures that they seek to represent. The thermal approach is also likely to naturally avoid inclusion of oscillatory gravity wave motions, which may contribute substantially to mass flux especially in stable regions of the atmosphere (Mrowiec et al., 2015)."

Additional references:

Zhao, M., & Austin, P. H. (2005). Life Cycle of Numerically Simulated Shallow Cumulus Clouds. Part II: Mixing Dynamics, Journal of the Atmospheric Sciences, 62(5), 1291-1310. https://doi.org/10.1175/JAS3415.1

Wu, J., A.D. Del Genio, M.-S. Yao, and A.B. Wolf, 2009: WRF and GISS SCM simulations of convective updraft properties during TWP-ICE. J. Geophys. Res., 114, D04206, doi:10.1029/2008JD010851.

*"Comments*

**R2C1:** *"The results and conclusions that are stressed most (use thermal framework for analyses; thermal framework yielding an abundance of additional information; thermals are dynamical and microphysical building blocks) are not well supported. If anything, most results are similar between the thermal framework and the cloudy updraft framework. Some are different, notably dynamics at upper levels, which is understandable given the low thresholds in the cloudy updraft framework that will pick up on detrained, buoyant air. How relevant these differences are for understanding or parameterizing convective clouds is not clear. It is simply stated that they are important with the thermal framework being superior, but what analyses support this? However, I don't believe that these are the most important conclusions anyway. I suggest shifting some of the focus to reflect the most important conclusions: (i) for liquid convective clouds, the thermal and cloudy updraft frameworks provide similar results (which is great since we don't have to disregard many past studies), (ii) for mixed phase and ice portions of convective clouds, substantial dynamical differences appear but microphysical sensitivities to aerosols remain similar, (iii) non-monotonic aerosol effects on liquid cloud updraft thermal number and buoyancy are seen, but no clear effects on the mixed phase and ice portions of updrafts are seen despite large sensitivities of cloud droplets to aerosol concentration. This last result is consistent with some recent studies showing warm phase invigoration without cold phase invigoration but goes against much of the aerosol deep convection invigoration studies concluding cold phase invigoration occurs, particularly in warm cloud base, isolated deep convection like this study examines."*

We thank this reviewer for this excellent suggestion, which we believe summarizes most of the other more specific comments in this review. Based on the various additions and modifications that derive from them, our main results and conclusions are now oriented in this direction, except in one respect: the "warm phase invigoration without cold phase invigoration". Although it seems plausible, we consider that our simple representation of cold microphysics is not enough in order to support this discussion (see comments **R2C3.4**, **R2C3.5** and **R2C3.6**). However, regarding points (i) and (ii), we have modified several parts of the text that follow this useful suggestion. Most of these additions or modifications follow from the general comment **R2GC** above, which touches on the main aspects of this idea, or from other specific comments below. For example, modified text at lines 12-13, 69-75 (73-79 in track changes), 281-282 (325-326 in track changes), 312-313 (357-358 in track changes), and added text starting at line 325 (lines 374-379 in track changes).

**R2C2:** *"There is a lot of subjective language and confusing terms used:*
*R2C2.1: Lines 42-43: "in which the dynamics of convection are resolved" is*
*ambiguous. What dynamics? The primary updraft or downdraft size, average*
*intensity, peak intensity? Many would not consider 250-m grid spacing sufficient to*
*resolve primary updrafts of many types of moist convection. Studies like Bryan et al.*
*(2003) and Lebo and Morrison (2015) show that 250 m is barely enough to resolve*
*the peak in the kinetic energy spectra, but those studies are also for continental*
*squall lines that may have larger, more intense updrafts than in other regimes such*
*as those in oceanic regions."*

To avoid this ambiguity, we changed lines 42-43 to "...in which cumulus convection does not require being parameterized...".

**R2C2.2:** *"Lines 47-48: Not all moist convection is necessarily constituted of short-lived thermals. Supercells, for example, can have 10-km wide plume updrafts with slab inflow layers. Morrison et al. (2020) and Peters et al. (2020) describe a thermal to plume spectrum dependent on updraft width and environmental conditions such as humidity, instability, and wind shear."*

We have rephrased lines 47-48 (49-50 in track changes version) to: "However, with notable exceptions as in supercells, moist convection commonly constitutes a series of many short-lived thermals…"

We now note a recent observational publication that finds a predominance of small updrafts in tropical convection (Yeung et al., 2021) starting at line 57 (lines 59-60 in track changes version): "For instance, recent observations by Yeung et al. (2021) indicate that most updrafts are less than 2 km deep, suggesting that a large fraction of mass flux may be left out by such selection criteria."

Yeung, N. K. H., Sherwood, S. C., Protat, A., Lane, T. P., & Williams, C. (2021). A Doppler Radar Study of Convective Draft Lengths over Darwin, Australia, Monthly Weather Review, 149(9), 2965-2974, https://doi.org/10.1175/MWR-D-20-0390.1

**R2C2.3:** *"Lines 70-72: I don't understand what it means for microphysical processes to be contained within thermals and driven by their internal circulations. This seems obvious that a microphysical process rate will depend on its local environment, whether advection, condensation, phase changes, or hydrometeor interactions."*

Indeed, this sentence describes something obvious, and what we should rather stress here is that the thermal framework produces additional information. We have removed the sentence referred to above, and modified this fragment as pointed out in comment **R2GC** (lines 73-81 in the track changes version).

**R2C2.4:** *"Line 73-74: What does it mean to be the basic dynamical entity of a cumulus cloud?"*

Since this sentence may be too vague here, we have removed it, as part of the

changes from the previous comment (**R2C2.3**) and the general comment **R2GC**.

**R2C2.5:** *"Line 89: Is this implying that the urban region heating is key for the sea breeze forcing initiating convection? A review of NEXRAD from this event shows convective precipitation initiating all along a sea breeze between Galveston, Houston, and Beaumont regardless of land cover with the most intense observed cells over rural locations in between Houston and Beaumont."*

We have removed "especially urban regions" from line 89 (99 in track changes).

**R2C2.6:** *"Line 115: Thermals are possible once resolution is sufficiently high, but that doesn't mean that these have been observed as is stated."*

We have changed "observed" to "expected" at line 115 (142 in track changes).

**R2C2.7:** *"Lines 142-143: Clarify what is meant here. Bryan et al. (2003) say that 250 m is sufficient for obtaining an inertial subrange, but this is also for a squall line and all results still do not converge at 125 m."*

We have removed this sentence (lines 172-173 in track changes).

**R2C2.8:** *"Line 162: I understand the thermal as an entity, but it seems overboard to call it a natural cloud chamber when it clearly has significant exchanges across its boundaries."*

We have removed this sentence (line 193 in track changes).

**R2C2.9:** *"Line 246: I don't understand why this suggests that thermals act as cloud chambers."*

We have removed this sentence too (line 286 in track changes).

**R2C2.10:** *"Line 318: What are "thermal microphysics quantities"?"*

Indeed, this was a typo. We have removed "thermal" from this phrase at line 318 (364 in track changes).

**R2C2.11:** *"Second to last sentence of abstract: Cumulus thermals can serve as a stronger foundation for improving sub-grid parameterizations than what? Which parameterizations? Why?"*

Traditional mass flux parameterizations use steady state assumptions (e.g., time integrated fluxes), whereas cumulus thermals are more directly connected to the supersaturation and cloud nucleation processes (i.e., updraft microphysics), which would be crucial information for new parameterization development. As mentioned toward the end of the reply to comment **R2GC**, we have added lines starting at line 325 (374-379 in track changes) to provide support to this sentence of the abstract.

**R2C2.12:** *"Last line of abstract: How do the result suggest that cumulus thermals are more realistic dynamical building blocks of cumulus convection and what are they more realistic than? What suggests that they are natural cloud chambers?"*

As mentioned in comment **R2GC**, we have removed this last sentence from lines 19-20.

**R2C3:** *"There are several results left unexplained or with interpretations not well supported by analyses."*

**R2C3.1**: *"Line 170: If supersaturation lowers, then condensation (and latent heating) increases, so what is compensating this extra latent heating to produce no net latent heating change? This should be explained."*

First of all, we have changed in line 169 (200 in track changes) "diabatic heating" for "latent heating". This is important because "latent heating" should not be considered as diabatic heating since it does not come from an external source. Thus, this phrasing is more appropriate.

Second, the latent heating term in our model simulations include all contributions due to phase changes of water; extracting these individual contributions would be required in order to investigate this. Since we do not have these separated terms available, we defer this to future work.

Clarification has been added starting at line 167 (199 in track changes): "... (summed over all source terms)", and at line 170 (lines 202-204 in track changes): "A possible explanation is that supersaturation differences are sustained within the context of negligibly different total condensate production rates within the thermal core, but that hypothesis cannot be definitively supported without additional diagnostics that separate the sources of latent heat in future work."

**R2C3.2:** *"Lines 183-185: How does the vertical wind speed profile highlight the importance of microphysical processes when its impact on microphysical processes aren't quantified?"*

Our point here is perhaps simpler (and maybe more obvious) than what the reviewer may have understood: when comparing both frameworks, higher average vertical velocity values aloft come together with higher values of microphysics quantities, which reflects the important coupling between dynamics and microphysics. To avoid this confusion, we have rephrased this sentence at lines 184-185 (218-219 in track changes): "This also results in a slightly more top-heaviness of profiles of other quantities, which reflects how strongly coupled are microphysical processes with updraft dynamics."

**R2C3.3**: *"It's not clear how robust (i.e., significant, which is a word that is used in the text) any inter-simulation differences are relative to variability expected from an ensemble with perturbed initial conditions. This is admitted by the authors – that it is difficult to discern a signal from the noise, but then the differences are described anyway as though they are robust."*

It is not clear to which descriptions exactly the reviewer refers to. Perhaps the

reviewer refers to lines 200-207 (240-247 in track changes), where we describe responses in terms of vertical velocity? We believe such a description is warranted, since many previous studies investigate such sensitivities in terms of "invigoration", and by doing so, we can actually illustrate more clearly the issue of signal-to-noise ratio, and the fact that individual pairs of experiments should be interpreted with care. Therefore, after describing these differences, we finish this paragraph with a clear warning (lines 206-207): "however, not all individual pairs of cases show such an increase, and the amplitude of the individual responses is usually larger than the average one."

Perhaps the reviewer refers to the description given in lines 215-230 (255-270 in track changes), where we mention that the response in buoyancy, w, DZ, mass flux, and number of tracked thermals appears to be "aerosol-limited".  In fact, this may provide at least a partial explanation for why individual pairs of experiments have different responses, so we consider that the fact that responses are different should not stop us from doing so. In fact, in the next comment the reviewer suggests us to investigate this further.

**R2C3.4**: *"There are different sensitivities to changes in aerosols depending on the magnitude of aerosol concentrations, but it isn't explained why this is and why changes are only visible for low-mid levels where presumably the thermals are dominated by liquid. I suggest reviewing previous studies on these topics."*

In these simulations, we have parameterized aerosol activation on cloud particles, and not yet parameterized ice nuclei impact. We now point this out starting at line 109 (lines 122-123 in track changes): "...while aerosol impact on ice nuclei is not considered."

In addition, this is single-ice P3 parameterization, and probably it is not sufficient to discuss much in overall aerosol impact yet without a 2nd class of ice category, if it is mixed with freshly nucleated ice mass and graupel-size ice mass. That is why this paper focuses on the dynamical and microphysical characteristics in the liquid phase. Of course, we would like to further develop the P3 scheme for a more robust ice microphysics process to steer our gear toward more robust aerosol-cloud interaction processes focusing on ice microphysics in a future study. But this study intends to characterize thermal microphysics states and their sensitivity as the first step. Therefore, we consider that investigating further these differences between low-mid levels and upper levels should be left for a future study.

**R2C3.5**: *"Line 245, 270-276: This also may be a result of larger regions of detrained, rising cloudy air at upper levels than at low levels, which could easily be examined. Since this is the largest difference between the two frameworks, an attempt at explaining it with a bit of investigation is warranted."*

Indeed, this is a possibility. However, following up on the previous comment (**R2C3.4**), in this study we are interested in the liquid phase, and our model setup is not ideal for a detailed investigation of such responses at upper levels. However, we have tested changing the vertical velocity threshold in the cloudy updraft definition to answer this and the following comment, which is closely related. Please refer to the reply to comment **R2C3.6** below for the complete response.

**R2C3.6**: *"Lines 281-282: These results show that a thermal framework produces some differences to the cloudy updraft framework, but it isn't clear why this implies their important role in cloud microphysics and dynamics. Clearly the most active portions of updrafts matter, but is the thermal definition needed for analyses of*

*updraft processes? The cloudy updraft definition is admittedly arbitrary and is a low bar for inclusion. If thresholds were increased, would results approach those of the thermal framework? The thermal framework rejects many updrafts. Does that influence interpretation of aerosol sensitivities?"*

As mentioned above in comment **R2GC**, we have modified this last sentence of the results section (now lines 325-326 in track changes), highlighting the added value of both frameworks.

Regarding the threshold for vertical velocity in the cloudy updraft definition, as pointed out in comment **R1C10**, the thresholds have been chosen so that both frameworks are as consistent as possible. However, the reviewer's suggestion to test higher thresholds in the cloudy updraft framework is a useful one. Below, Figure R8 is identical to Figure 7 in the manuscript, with cloudy updrafts defined with w>1m/s (reproduced here to facilitate comparison), while Figures R9 and R10 are obtained using higher thresholds of vertical velocity (w>2m/s and w>4m/s), and will be now included as supplementary material (Figures S4 and S5).

[Figure]

Fig. R8: Same as Figure 7 in the manuscript (with w>1m/s for cloudy updraft grid point definition), shown here for comparison to figures R9 and R10.

[Figure]

Fig. R9: Same as Figure 7 in the manuscript, but using w>2m/s for cloudy updraft grid point definition.

[Figure]

Fig. R10: Same as Figure 7 in the manuscript, but using w>4m/s for cloudy updraft grid point definition.

By increasing this vertical velocity threshold, some values, particularly at upper levels, do approach each other between frameworks. However, at the same time they depart from each other at middle and lower levels (e.g., latent heating rates, vertical velocity, nucleation rates), and by doing so they often make column-integrated values also less consistent between frameworks. There is clearly no unique w threshold for which all quantities agree in both frameworks at different levels. The effect of this threshold increase is most obvious in terms of vertical velocity (Figs. R8d-R10d), where it becomes clear that the differences between the two frameworks are not linear in terms of this threshold. So changing this parameter will not bring results from both frameworks together overall. In fact, based on Figures R8d-R10d, we argue that the most consistent comparison between the two frameworks is when the original threshold of w>1m/s for cloudy updraft grid points is used. Furthermore, differences between frameworks at upper levels should be further investigated with a better ice-microphysics representation. Finally, notice that aerosol sensitivities, which in these figures are evident from the relative changes between column integrated values (large crosses), are not affected by the higher w thresholds used in figures R9 and R10.

To summarize all this, we have added the following text starting at line 276 (lines 316-320 in track changes), as well as Figures R9 and R10 as supplementary material:

> "Increasing the vertical velocity threshold for the cloudy updraft grid point definition, while it does not modify the aerosol sensitivities found here, yields closer values between frameworks for several quantities at upper levels, but at the expense of larger differences at middle and lower levels that result in less overall consistency (Figs. S4 and S5). Further investigation of detailed differences between the two frameworks at upper levels is left for a future study, with a focus extended to ice microphysical processes."

**R2C4**: *"How are aerosols initialized in the free troposphere? If they are removed through deposition, how are they replenished?"*

The aerosol vertical profiles are set consistently to ACPC MIP Mode 1 (Fig R11, Marinescu et al. 2021). We have added the following sentences after line 109 (lines 123-126 in track changes, see also **R1C2**):

> "The polluted and clean aerosol size distributions and vertical profiles were based on the data from Deriving Information on Surface conditions from Column and Vertically

Resolved Observations Relevant to Air Quality (DISCOVER-AQ) in September 2013, as well as satellite-based estimates (Rosenfeld et al. 2012) near Houston on 19 June 2013. Timing of satellite CCN observations are identical to the simulation dates."

Followed by another addition (lines 126-133 in track changes):

"The profiles feature constant values in the boundary layer up to 2.5km and in the free troposphere over 5km with a linear transition between these heights. Aerosol removal/replenishment processes are based on semi-diagnostic methods in Fridlind et al. (2017). This method activates cloud droplets for a given supersaturation rate and aerosol characteristics, and tracks the sum of activated and unactivated aerosol through advection and mixing. Additional cloud droplets can be activated when newly activated cloud droplets number exceeds the present number of cloud droplets. Aerosol number concentrations will be reduced only when cloud droplets are reduced by coalescence process (i.e., autoconversion to precipitation class). The advantage of this approach is to account for activation/regeneration of aerosols without explicitly accounting for aerosols within cloud droplets (see details in Fridlind et al. 2017)."

[Figure]

Fig R11. Vertical profiles of initial aerosol concentrations. Unit of x-axis is #/cm$^3$.

"References

Bryan, G. H., Wyngaard, J. C., and Fritsch, J. M., 2003: Resolution Requirements for the Simulation of Deep Moist Convection, Mon. Wea. Rev., 131, 2394–2416, https://doi.org/10.1175/1520-0493(2003)131%3C2394:RRFTSO%3E2.0.CO;2.

Lebo, Z. J., and Morrison, H., 2015: Effects of Horizontal and Vertical Grid Spacing on Mixing in Simulated Squall Lines and Implications for Convective Strength and Structure. Mon. Wea. Rev., 143, 4355-4375, https://doi.org/10.1175/MWR-D-15-0154.1.

Morrison, H., Peters, J. M., Varble, A. C., Hannah, W. M., and Giangrande, S. E., 2020: Thermal Chains and Entrainment in Cumulus Updrafts. Part I: Theoretical Description, J. Atmos. Sci., 77, 3637–3660, https://doi.org/10.1175/JAS-D-19-0243.1.

Peters, J. M., Morrison, H., Varble, A. C., Hannah, W. M., and Giangrande, S. E., 2020: Thermal Chains and Entrainment in Cumulus Updrafts. Part II: Analysis of Idealized Simulations, J. Atmos. Sci., 77, 3661–3681, https://doi.org/10.1175/JAS-D-19-0244.1."

**Reply to Anonymous Referee #3 of the manuscript**

**"Updraft dynamics and microphysics: on the added value of the cumulus thermal reference frame in simulations of aerosol-deep convection interactions"**
by
Daniel Hernández-Deckers, Toshihisa Matsui, and Ann M. Fridlind

In the following we provide a point by point response to comments by Reviewer #3, where we quote in italics the original comments. We have numbered all comments in boldface, based on the reviewer number and comment number so that the different replies can be easily referred to within the text (e.g., R2C3 refers to comment number 3 from Reviewer #2).

*"Review of "Updraft dynamics and microphysics: on the added value of the cumulus thermal reference frame in simulations of aerosol-deep convection interactions"*

*Model-based analysis of aerosol indirect effects, or other properties of convection in general, typically relies on some definition of an updraft. Often this is done by considering all cloudy points in a model with some threshold vertical velocity value. The authors here use object tracking code to follow individual thermals, offering this as an alternative to the cloudy grid points method. They use this method to investigate some simple aerosol indirect effects in simulations of deep convection. Generally the aerosol effects on the warm part of the storms are as expected, and are fairly consistent between the two methods, but some differences are seen in the upper levels that suggest the thermal tracking method could be a useful way to investigate processes in deep convection.*

*Overall I think the paper is interesting, novel, and sound, but I offer some comments and suggestions below to help improve and clarify the discussions."*

We thank reviewer #3 for these useful comments and suggestions, which we reply to in the following text:

*"Comments:"*

**R3C1:** *"It seems to me that the authors overstate the importance of their method compared to the traditional approach. There is no "correct" answer in how to do the analysis. Both methods may prove useful for examining different characteristics or different types or regions of storms. It's especially concerning to me that the authors concentrate on the differences in the upper levels, yet this study doesn't at all investigate the ice phase. It's difficult to conclude anything about microphysics in the upper levels of a storm if only cloud and rain are included in the analysis. This merits some mention at least. What implications might the differences between these two approaches offer when examining the ice phase?"*

We agree that we may have overstated the importance of the thermal framework, and refer to the general comment from reviewer #2 (**R2GC**), where we deal with this issue. Regarding the differences found in the upper levels, we agree that since we cannot really investigate

the ice phase in this study, it is not possible to reach any conclusions regarding the microphysics. In fact, reviewer #2 suggested we investigate this further in comment **R2C3.4**, but we do not think this is warranted given the lack of robust ice microphysics processes in these simulations. Thus, here we can only describe certain dynamical responses we find at upper levels, but a detailed examination of the ice phase and its implications in terms of the two approaches must be left for a future study.

**R3C2:** *"Line 170: It doesn't seem consistent that there could be huge increases in nucleation rates, but not in latent heating. Is this being balanced by evaporation?"*

As now noted per response to comment **R2C3.1** and in added text in lines 19, one possible explanation is that sustained supersaturation within thermal cores exists within the context of relatively unchanged vapor condensation rates, and confirming that hypothesis would motivate reporting diabatic heating source contributions in future work.

**R3C3:** *"Line 172: I think it would be better to show at least some of this information. I found myself wondering about the properties of the thermals and the variability of those properties at multiple times while reading, and was frustrated to keep seeing "not shown"."*

Reading again through this part of the text we notice that the way it is written indeed gives the impression that there are many important results that we do not show. However, this concerns 2 types of plots: first, thermal's composite lifetime, vertical distance traveled (DZ), and radius; and second, histograms of these quantities. Except for the thermal's lifetime, notice that Fig. 6 shows vertical profiles of these quantities, which provides even more detailed information than the simple average composites, since the latter would be averaged over vertical levels. So in fact, instead of saying "not shown", we can refer to these plots instead. We have modified this sentence at line 172 to (lines 205-207 in the track changes version):

> "We do not find any prominent trends in terms of the thermals' composite lifetime, vertical distance traveled (DZ) or radius (R). For R and DZ, this can be inferred from the vertical profiles shown in Figs. 6b,d."

Regarding the histograms, we include them here for the review process (Fig. R12), but we do not think that an additional figure in the manuscript is warranted. As mentioned in the paper, they do not show any important changes when aerosol concentrations are increased.

[Figure]

Fig. R12: Normalized distributions (their integrals must equal unity) of a) thermals' radius R, b) vertical distance traveled by thermals DZ, and c) thermals' lifetime in the experiments with successive doubling of aerosol concentrations (see legend in top panel, in $cm^{-3}$).

**R3C4:** *"Line 188: While it is likely true that the convective core is more tightly linked at upper levels to fewer, stronger updrafts, the fact remains that the additional updrafts not captured as thermals do exist there, and are certainly quite relevant for microphysics. By only looking at the strongest updrafts in the upper levels, it may better capture that core, but is ignoring stratiform processes. It's not clear that one of these methods would be better than the other in general - it depends on the question being asked. "*

We agree, and this comes back to comment **R3C1**, and to the general comment **R2GC** from reviewer #2. Please refer to our replies to those comments, which address this issue.

**R3C5:** *"Line 210: I'm not sure that I buy that it's representative of the total mass flux. In figure 4 you show that the profiles have different shapes as well as the difference in magnitude. They are not really looking at the same thing, which is your point, so how is one representative of the other?"*

Thank you for pointing this out. The problem here is that the "total mass flux" can be a rather subjective quantity, because it depends on the sampling strategy. We have rephrased this to make it consistent with what Hernandez-Deckers and Sherwood (2016) show, which is that it

is representative of the "entire convective activity" (see modified text at lines 250-251 in the track changes version).

**R3C6:** *"Figure 6: How are there 300 individual thermals in this fairly small domain?"*

We realize now that we had not mentioned explicitly the size of this sampling domain where thermals are tracked (100x100km), but only showed it graphically in Fig. 1. We now do this at line 113 (lines 139-140 of the track changes version). Notice that this is actually a large area compared to the typical size of thermals. Furthermore, these numbers are accumulated over a 3-hour period in each simulation, so it is perfectly possible to have 300 thermals in this volume. Actually, this is most likely only a fraction of the "actual" number of thermals.

**R3C7:** *"Figure 6: With 250m grid spacing, an average radius of 500m doesn't make a lot of sense. These thermals are barely resolved, and this being the average means you are also tracking thermals that are fewer than 4 points across."*

By construction, the thermal tracking algorithm does not track any thermal with a radius smaller than twice the horizontal grid spacing, so no tracked thermal has a radius under 500m. Notice from Fig. 6b that only at the lowest layer below z=1km (where only a handful of thermals are tracked), does the average radius reach values below 1000m, but never 500m. Above this level, where most thermals are tracked, it is always larger than ~1000m, thus at least 8 or more grid points across a thermal's diameter. In fact, the medians (which are lower than the means) of the distributions in Fig. R9 are 1020m, 1031m, 1018m and 990m, and more than ~85% of the thermals have radii larger than 750m in all simulations, thus more than 6 grid points across.

In order to make this clear to the reader, we have added the following text to the manuscript after line 125 (lines 154-155 in the track changes version):
> "The smallest radius permitted for a thermal is twice the model grid spacing, thus 500m in this case. Smaller thermals are discarded."